# Offline Inverse Constrained Reinforcement Learning for Safe-Critical Decision Making in Healthcare

## Abstract

Reinforcement Learning (RL) applied in healthcare can lead to unsafe medical decisions and treatment, such as excessive dosages or abrupt changes, often due to agents overlooking common-sense constraints. Consequently, Constrained Reinforcement Learning (CRL) is a natural choice for safe decisions. However, specifying the exact cost function is inherently difficult in healthcare. Recent Inverse Constrained Reinforcement Learning (ICRL) is a promising approach that infers constraints from expert demonstrations. ICRL algorithms model Markovian decisions in an interactive environment. These settings do not align with the practical requirement of a decision-making system in healthcare, where decisions rely on historical treatment recorded in an offline dataset. To tackle these issues, we propose the Constraint Transformer (CT). Specifically, 1) utilize causal attention mechanism to incorporate historical decisions and observations into the constraint modeling and employ a non-Markovian layer for weighted constraints to capture critical states, 2) generative world model to perform exploratory data augmentation, thereby enabling offline RL methods to generate unsafe decision sequences. In multiple medical scenarios, empirical results demonstrate that CT can capture unsafe states and achieve strategies that approximate lower mortality rates, reducing the occurrence probability of unsafe behaviors.

## 1 Introduction

In recent years, the doctor-to-patient ratio imbalance has drawn attention, with the U.S. having only 223.1 physicians per 100,000 people [1]. AI-assisted therapy emerges as a promising solution, offering timely diagnosis, personalized care, and reducing dependence on experienced physicians. Therefore, the development of an effective AI healthcare assistant is crucial.

Reinforcement learning (RL) offers a promising approach to develop AI assistants by addressing sequential decision-making tasks. However, this method can still lead to unsafe behaviors, such as administering excessive drug dosages, inappropriate adjustments of medical parameters, or abrupt changes in medication dosages. These behaviors, such as **"too high"** or **"sudden change"** can significantly endanger patients, potentially resulting in acute hypotension, hypertension, arrhythmias, and organ damage, with fatal consequences [4, 5, 6]. For example, in sepsis treatment, patients receiving vasopressors (vaso) at dosages exceeding $1\mu g/(kg \cdot min)$ have a mortality rate of $90\%$ [7]. Moreover, the **"sudden change"** in vaso can rapidly affect blood vessels, causing acute fluctuations in blood pressure and posing life-threatening risks to patients [8]. Our experiments demonstrate that the work

Table 1: The proportion of unsafe behaviors occurrences in vaso suggested by physician and DDPG. The typical range for vaso is $0.1 \sim 0.2\mu g/(kg \cdot min)$, with doses exceeding $0.5$ considered high [2]. A cutoff value of $0.75$ is identified as a critical threshold associated with increased mortality [3].

| Drug dosage ($\mu g/(kg \cdot min)$) | Physician | DDPG |
|---|---|---|
| vaso > 0.75 | 2.27% | 7.44% ↑ |
| vaso > 0.9 | 1.71% | 7.40% ↑ |
| $\Delta$ vaso > 0.75 | 2.45% | 21.00% ↑ |
| $\Delta$ vaso > 0.9 | 1.88% | 20.62% ↑ |

$\Delta$ vaso: The change in vaso between two-time points.

[9] applying the Deep Deterministic Policy Gradient (DDPG) algorithm in sepsis indeed exhibits **"too high"** and **"sudden change"** [1] unsafe behaviors in vaso recommendations, as shown in Table 1.

This paper aims to achieve safe healthcare policy learning to mitigate unsafe behaviors. The most common method for learning safe policies is Constrained Reinforcement Learning (CRL) [10, 11], with the key to its success lying in the constraints representation. However, in healthcare, we can only design the cost function based on prior knowledge, which limits its application due to a lack of personalization, universality, and reliance on prior knowledge. For more details about issues, please refer to Appendix A. Therefore, Inverse Constrained Reinforcement Learning (ICRL) [12] emerges as a promising approach, as it can infer the constraints adhered to by experts from their demonstrations. However, directly applying ICRL in healthcare presents several challenges:

**1) The Markov decision is not compatible with medical decisions.** ICRL algorithms model Markov decisions, where the next state depends only on the current state and not on the history [13, 14]. However, in healthcare, the historical states of patients are crucial for medical decision-making [15], as demonstrated in the experiments shown in Figure 1. Therefore, ICRL algorithms based on Markov assumption can not capture patient history, and ignore individual patient differences, thereby limiting effectiveness.

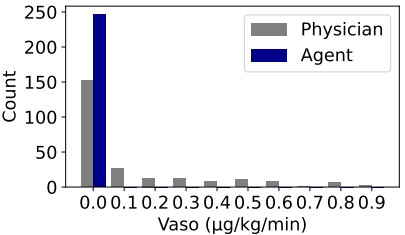

**2) Interactive environment is not available for healthcare or medical decisions.** ICRL algorithms [12, 16] follow an online learning paradigm, allowing agents to explore and learn from interactive environments. However, unrestricted exploration in healthcare often entails unsafe behaviors that could breach constraints and result in substantial losses. Therefore, it is necessary to infer constraints using only offline datasets.

Figure 1: The distribution of vaso for patients with the same state. The physician makes different decisions due to referencing historical information, while the agent based on Markov decision-making can only make the same decision.

In this paper, we introduce offline Constraint Transformer (CT), a novel ICRL framework that incorporates patients' historical information into constraint modeling and learns from offline data to infer constraints in healthcare. Specifically,

1) Inspired by the recent success of transformers in sequence modeling [17, 18, 19], we incorporate historical decisions and observations into constraint modeling using a causal attention mechanism. To capture key events in trajectories, we introduce a non-Markovian transformer to generate constraints and importance weights, and then define constraints using weighted sums. CT takes trajectories as input, allowing for the observation of patients' historical information and evaluation of key states.

2) To learn from an offline dataset, we introduce a model-based offline RL method that simultaneously learns a policy model and a generative world model via auto-regressive imitation of the actions and observations in medical decisions. The policy model employs a stochastic policy with entropy regularization to prevent it from overfitting and improve its robustness. Utilizing expert datasets, the generative world model uses an auto-regressive exploration generation paradigm to effectively discover a set of violating trajectories. Then, CT can infer constraints in healthcare through these unsafe trajectories and expert trajectories.

In the medical scenarios of sepsis and mechanical ventilation, we conduct experimental evaluations of offline CT. Experimental evaluations demonstrate that offline CT can capture patients' unsafe states and assign higher penalties, thereby providing more interpretable constraints compared to previous works [9, 20, 21]. Compared to unconstrained and custom constraints, CT achieves strategies that closely approximate lower mortality rates with a higher probability (improving by $8.85\%$ compared to DDPG). To investigate the avoidance of unsafe behaviors with offline CT, we evaluate the probabilities of "too high" and "sudden changes" occurring in the sepsis. The experimental results show that CRL with CT can reduce the probability of unsafe behaviors to zero.

## 2   Related Works

**Reinforcement Learning in Healthcare.** RL has made great progress in the realm of healthcare, such as sepsis treatment [9, 20, 21, 22], mechanical ventilation [23, 24, 25], sedation [26] and anesthesia

---

[1]In sepsis, "too high" indicates that the dosage of the vaso medication exceeds the threshold. And "sudden change" indicates that the change in vaso medication dosage between two time points exceeds the threshold.

[27, 28]. However, these works mentioned above have not addressed potential safety issues such as sudden changes or too high doses of medication. Therefore, the development of policies that are both safe and applicable across various healthcare domains is crucial.

**Inverse Constrained Reinforcement Learning.** Previous works inferred constraint functions by determining the feasibility of actions under current states. In discrete state-action spaces, Chou *et al.* [29] and Park *et al.* [30] learned constraint sets to differentiate constrained state-action pairs. Scobee & Sastry [31] proposed inferring constraint sets based on the principle of maximum entropy, while some studies [32, 33] extended this approach to stochastic environments using maximum causal entropy [34]. In continuous domains, Malik *et al.* [12], Gaurav *et al.* [16], and Qiao *et al.* [35] used neural networks to approximate constraints. Some works [11, 29] applied Bayesian Monte Carlo and variational inference to infer the posterior distribution of constraints in high-dimensional state spaces. Xu *et al.* [36] modeled uncertainty perception constraints for arbitrary and epistemic uncertainties. However, these methods can only be applied online and lack historical dependency.

**Transformers for Reinforcement Learning.** Transformer has produced exciting progress on RL sequential decision problems [17, 18, 37, 38]. These works no longer explicitly learn Q-functions or policy gradients, but focus on action sequence prediction models driven by target rewards. Chen *et al.* [18] and Janner *et al.* [37] perform auto-regressive modeling of trajectories to achieve policy learning in an offline environment. Furthermore, Zheng *et al.* [17] unify offline pretraining and online fine-tuning within the Transformer framework. Liu *et al.* [38] and Kim *et al.* [19] integrate the transformer architecture into constraint learning and preference learning. The transformer architecture, with its sequence modeling capability and independence from the Markov assumption, can capture temporal dependencies in medical decision-making. Thus, it is well-suited for trajectory learning and personalized learning in medical settings.

## 3 Problem Formulation

We model the medical environment with a Constrained Markov Decision Process (CMDP) $\mathcal{M}^c$ [39], which can be defined by a tuple $(\mathcal{S}, \mathcal{A}, \mathcal{P}, \mathcal{R}, \mathcal{C}, \gamma, \kappa, \rho_0)$. Similar to studies [23, 40], we extract data within 72 hours of patient admission, with each 4-hour interval constituting a window or time step. The state indicators of the patient at each time step are denoted as $s \in \mathcal{S}$. The administered drug doses or instrument parameters of interest are considered as actions $a \in \mathcal{A}$, while reward function $\mathcal{R}$ is used to describe the quality of the patient's condition and provided by experts based on prior work [9, 23]. At each time step $t$, an agent performs an action $a_t$ at a patient's state $s_t$. This process generates the reward $r_t \sim \mathcal{R}(s_t, a_t)$, the cost $c_t \sim \mathcal{C}$ and the next state $s_{t+1} \sim \mathcal{P}(\cdot \mid s_t, a_t)$, where $\mathcal{P}$ defines the transition probabilities. $\gamma$ denotes the discount factor. $\kappa \in \mathbb{R}_+$ denotes the bound of cumulative costs. $\rho_0$ defines the initial state distribution. The goal of the CRL policy $\pi$ is to maximize the reward return while limiting the cost in a threshold $\kappa$:

$$\arg \max_{\pi} \mathbb{E}_{\pi, \rho_0}[\sum_{t=1}^{T} \gamma^t r_t], \quad \text{s.t.} \quad \mathbb{E}_{\pi, \rho_0}[\sum_{t=1}^{T} \gamma^t c_t] \leq \kappa. \tag{1}$$

where $T$ is the length of the trajectory $\tau$. CRL commonly assumes that constraint signals are directly observable. However, in healthcare, such signals are not easily obtainable. Therefore, Our objective is to infer reasonable constraints for CRL to achieve safe policy learning in healthcare.

**Safe-Critical Decision Making with Constraint Inference in Healthcare.** Our general goal is for our policy to approximate the optimal policy, which refers to the strategy under which the patient's mortality rate is minimized (achieving a zero mortality rate is often difficult since there are patients who can not recover, regardless of all potential future treatment sequences [41]). Decision-making with constraints can formulate safer strategies by discovering and avoiding unsafe states, thereby approaching the optimal policy.

However, most offline RL algorithms rely on online evaluation, where the agent is evaluated in an interactive environment, whereas in medical scenarios, only offline evaluation can be utilized. In previous works [5, 9, 40, 42], they qualitatively analyzed by comparing the differences (DIFF) between the drug dosage recommended by our policy $\pi$ and the dosage administered by clinical physicians $\hat{\pi}$, and its relationship with mortality rates, through graphical analysis. In the graph depicting the relationship between the DIFF and mortality rate, at the point when DIFF is zero, the lower the mortality rate of patients, the better the performance of the policy [40]. To provide a more accurate quantitative evaluation, we introduce the concept of the probability of approaching the optimal policy, defined as $\omega$:

$$\omega = \frac{\text{Number of survivors among the top } N \text{ patients}}{N} \tag{2}$$

We randomly collect $2N$ patients (with an equal number of known survivors and non-survivors under doctor's policy $\hat{\pi}$) from the offline dataset. We then calculate the DIFF and sort it in ascending order. The optimality of the policy can be evaluated through the following two points: 1) The higher the survival probability (i.e., $\omega$) of the top $N$ patients, the lower the mortality rate can be achieved by executing $\pi$; 2) The smaller the DIFF among the surviving patients in the top $N$, the greater the probability that $\pi$ is optimal.

## 4 Method

To infer constraints and achieve safe decision-making in healthcare, we introduce the Offline Constraint Transformer (Figure 2), a novel ICRL framework.

**Inverse Constrained Reinforcement Learning.** ICRL aims to recover the cost function $\mathcal{C}^*$ by leveraging a set of trajectories $\mathcal{D}_e = \{\tau_e^{(i)}\}_i^N$ sampled from an expert policy $\pi_e$, where $N$ denotes the number of the trajectories. ICRL is commonly based on the Maximum Entropy framework [31], and the likelihood function is articulated as [12]:

$$p(\mathcal{D}_e \mid \mathcal{C}) = \frac{1}{(Z_{\mathcal{M}^c})^N} \prod_{i=1}^{N} \exp\left[R(\tau^{(i)})\right] \mathbb{I}^{\mathcal{M}^c}(\tau^{(i)}) \tag{3}$$

Here, $Z_{\mathcal{M}} = \int \exp(\beta r(\tau)) \mathbb{I}^{\mathcal{M}}(\tau) d\tau$ is the normalizing term. The indicator $\mathbb{I}^{\mathcal{M}^c}(\tau^{(i)})$ signifies the extent to which the trajectory $\tau^{(i)}$ satisfies the constraints. It can be approximated using a neural network $\zeta_\theta(\tau^{(i)})$ parameterized with $\theta$, defined as $\zeta_\theta(\tau^{(i)}) = \prod_{t=0}^{T} \zeta_\theta(s_t^i, a_t^i)$. Consequently, the cost function can be formulated as $C_\theta = 1 - \zeta_\theta$. Substituting the neural network for the indicator, we can update $\theta$ through the gradient of the log-likelihood function:

$$\nabla_\theta \mathcal{L}(\theta) = \mathbb{E}_{\tau^{(i)} \sim \pi_e} \left[\nabla_\theta \log[\zeta_\theta(\tau^{(i)})]\right] - \mathbb{E}_{\hat{\tau} \sim \pi_{\mathcal{M}^{\hat{\zeta}_\theta}}} \left[\nabla_\theta \log[\zeta_\theta(\hat{\tau}^{(i)})]\right] \tag{4}$$

where $\mathcal{M}^{\hat{\zeta}_\theta}$ denotes the MDP obtained after augmenting $\mathcal{M}$ with the cost function $C_\theta$, using the executing policy $\pi_{\mathcal{M}^{\hat{\zeta}_\theta}}$. And $\hat{\tau}$ are sampled from the policy. In practice, ICRL can be conceptualized as a bi-level optimization task [11]. We can 1) update this policy based on Equation 1, and 2) employ Equation 4 for constraint learning. Intuitively, the objective of Equation 4 is to distinguish between trajectories generated by expert policies and imitation policies that may violate the constraints.

Specifically, task 1) involves updating the policy using advanced CRL methods. Significant progress has been made in some works such as BCQ-Lagrangian (BCQ-Lag), COpiDICE [43], VOCE [44], and CDT [38]. Meanwhile, task 2) focuses on learning the constraint function, as shown in Figure 2. Our research primarily improves the latter process due to two main challenges facing ICRL in healthcare: **Challenge 1)** pertains to the limitations of the Markov property, and **Challenge 2)** involves the issue of inferring constraints only from offline datasets. To address these challenges, we propose the offline CT as our solution.

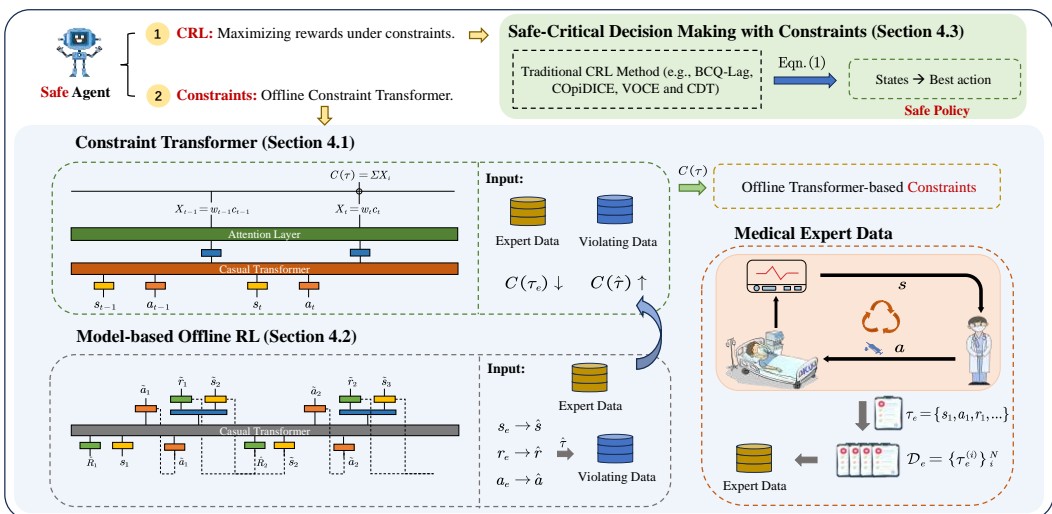

Figure 2: The overview of the safe healthcare policy learning with offline CT.

174 **Offline Constraint Transformer.** To address the first challenge, we delve into the inherent issues of
175 applying the Markov property to healthcare and draw inspiration from the successes of Transformer
176 in decision-making, redefining the representation of the constraints. To realize the offline training, we
177 consider the essence of ICRL updates, proposing a model-based RL to generate unsafe behaviors
178 used to train CT. We outline three parts: establishing the constraint representation model (Section
179 4.1), creating an offline RL for violating data (Section 4.2), and learning safe policies (Section 4.3).

## 180 4.1 Constraint Transformer

181 ICRL methods relying on the Markov prop-
182 erty overlook patients' historical informa-
183 tion, focusing only on the current state.
184 However, both current and historical states,
185 along with vital sign changes are crucial
186 for a human doctor's decision-making pro-
187 cess [15]. To emulate the observational
188 approach of humans, we draw inspiration
189 from the Decision Transformer (DT) [18]
190 to incorporate historical information into
191 constraints for a more comprehensive ob-
192 servation and judgment. We propose a
193 constraint modeling approach based on a

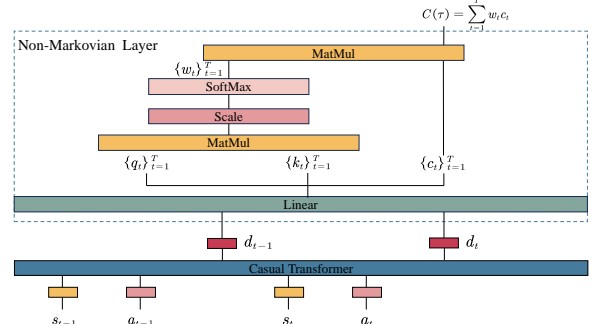

Figure 3: The structure of the Constraint Transformer.

194 causal attention mechanism, as shown in Figure 3. The structure comprises a causal Transformer for
195 sequential modeling and a non-Markovian layer for weighted constraints learning.

196 **Sequential Modeling for Constraints Inference.** For a trajectory segment of length $T$, $2T$ input
197 embeddings are generated, with each position containing state $s$ and action $a$ embeddings. Addi-
198 tionally, these embeddings undergo linear and normalization layers before being fed into the causal
199 Transformer, which produces output embeddings $\{d_t\}_{t=1}^T$ determined by preceding input embeddings
200 from $(s_1, a_1, ..., s_T, a_T)$. Here, $d_t$ depends only on the previous $t$ states and actions.

201 **Modeling Non-Markovian for Weighted Constraints Learning.** Although $d_t$ represents the cost
202 function $c_t$ derived from observations over long trajectories, it doesn't pinpoint which previous key
203 actions or states led to its increase. In healthcare, identifying key actions or states is vital for analyzing
204 risky behaviors and status, and enhancing model interpretability. To address this, we draw inspiration
205 from the design of the preference attention layer in [19] and introduce an additional attention layer.
206 This layer is employed to define the cost weight for non-Markovians. It takes the output embeddings
207 from the causality transformer as input and generates the corresponding cost and importance weights.
208 The output of the attention layer is computed by weighting the values through the normalized dot
209 product between the query and other keys:

$$\sum_{t=1}^T \text{softmax}\left(\{\langle q_t, k_{t'}\rangle\}_{t'=1}^T\right)_t \cdot c_t = \sum_{t=1}^T w_t \cdot c_t \tag{5}$$

210 Here, the key $k_t \in \mathbb{R}^m$, query $q_t \in \mathbb{R}^m$, and value $c_t \in \mathbb{R}^m$ are derived from the $t$-th input $d_t$
211 through linear transformations, where $m$ denotes the embedding dimension. Furthermore, for each
212 time step $t$, since $d_t$ depends only on the previous state-action pairs $\{(s_i, a_i)\}_{i=1}^t$ and serves as the
213 input embedding for the attention layer, $c_t$ is also associated solely with the preceding $t$ time steps.
214 The representation of the cost function as a weighted sum is defined as $C(\tau) = \sum_{t=1}^T w_t \cdot c_t$. Then,
215 we can also determine the constraint function values for each preceding subsequence. Introducing the
216 newly defined cost function, we redefine Equation 4 for CT as:

$$\nabla_\phi \mathcal{L}(\phi) = \mathbb{E}_{\hat{\tau} \sim \mathcal{D}_v}\left[\nabla_\phi \log[C_\phi(\hat{\tau})]\right] - \mathbb{E}_{\tau \sim \mathcal{D}_e}\left[\nabla_\phi \log[C_\phi(\tau)]\right] \tag{6}$$

217 where $\phi$ is the parameter of CT, $\mathcal{D}_e$ and $\mathcal{D}_v$ represent the expert data and the violating data. This
218 formulation implies that the constraint should be minimized on the expert policy and maximized on
219 the violating policy. We construct an expert and a violating dataset to evaluate Equation 6 in offline.
220 The expert data can be acquired from existing medical datasets or hospitals. Regarding the violating
221 dataset, we introduce a generative model to establish it, as detailed in Section 4.2.

## 222 4.2 Model-based Offline RL

To train CT offline, we introduce a model-based offline RL method (Figure 4) to generate violating data that refers to unsafe behavioral data and can be represented as $\tau_v = (s_1, a_1, r_1, s_2, ...) \in \mathcal{D}_v$. The model simultaneously learns a policy model and a

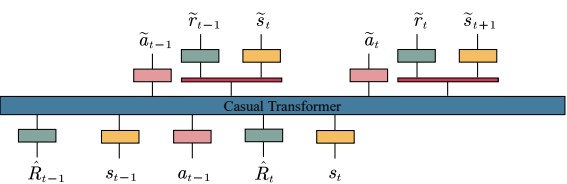

Figure 4: The structure of the model-based offline RL.

generative world model via auto-regressive imitation of the actions and observations in healthcare. The model processes a trajectory, $\tau_e \in \mathcal{D}_e$, as a sequence of tokens encompassing the return-to-go, states, and actions, defined as $(\hat{R}_1, s_1, a_1, ..., \hat{R}_T, s_T, a_T)$. Notably, the return-to-go $\hat{R}_t$ at timestep $t$ is the sum of future rewards, calculated as $\hat{R}_t = \sum_{t'=t}^{T} r_{t'}$. At each timestep $t$, it employs the tokens from the preceding $K$ timesteps as its input, where $K$ represents the context length. Thus, the input tokens for it at timestep $t$ are denoted as $h_t = \{\hat{R}_{-K:t}, s_{-K:t}, a_{-K:t-1}\}$, where $\hat{R}_{-K:t} = \{\hat{R}_K, ..., \hat{R}_t\}$, $s_{-K:t} = \{s_K, ..., s_t\}$ and $a_{-K:t-1} = \{a_K, ..., a_{t-1}\}$.

**Policy Model.** The input tokens are encoded through a linear layer for each modality. Subsequently, the encoded tokens pass through a casual transformer to predict future action tokens. We use a stochastic policy [38] to achieve policy learning. Additionally, we utilize a Shannon entropy regularizer $\mathcal{H}\left[\pi_\vartheta(\cdot \mid h)\right]$ to prevent policy overfitting and enhance robustness. The optimization objective is to minimize the negative log-likelihood loss while maximizing the entropy with weight $\lambda$:

$$\min_\vartheta \quad \mathbb{E}_{h_t \sim \mathcal{D}_e}[-\log \pi_\vartheta(\cdot \mid h_t) - \lambda\mathcal{H}\left[\pi_\vartheta(\cdot \mid h_t)\right]] \tag{7}$$

where the policy $\pi_\vartheta(\cdot \mid h_t) = \mathcal{N}(\mu_\vartheta(h_t), \Sigma_\vartheta(h_t))$ adopts the stochastic Gaussian policy representation and $\vartheta$ is the parameter.

**Generative World Model.** To predict states and rewards, we use $x_t = \{h_t \cup a_t\}$ as input encoded by linear layers. The encoded tokens pass through the casual transformer to predict hidden tokens. Then we utilize two linear layers to fit the rewards and states. The optimization objective for the two linear layers $\ell$ with the parameters $\varphi$ and $\mu$ can be defined as:

$$\min_{\varphi, \mu} \quad \mathbb{E}_{s_t, r_{t-1} \in x_t \sim \mathcal{D}_e}[(s_t - \ell_\varphi(x_t))^2 + (r_{t-1} - \ell_\mu(x_t))^2] \tag{8}$$

**Generating Violating Data.** In RL, excessively high rewards, surpassing those provided by domain experts, may incentivize agents to violate the constraints in order to maximize the total reward [11]. Therefore, we set a high initial target reward $\hat{R}_1$ to obtain violation data. We feed $\hat{R}_1$ and initial state $s_1^{(i)}$ into the model-based offline RL to generate $\tau_v^{(i)}$ in an auto-regressive manner, as depicted in model-based offline RL of Figure 2, where $\tilde{a}$, $\tilde{r}$ and $\tilde{s}$ are predicted by the model. The target reward $\hat{R}$ decreases incrementally and can be represented as $\hat{R}_{t+1} = \hat{R}_t - \tilde{r}_t$. Considering the average error in trajectory prediction, we generate trajectories with the length $K = 10$, as detailed in Appendix B.3. Repeating $N$ initial states, we can get violating data $\mathcal{D}_v = \{\tau_v^{(i)}\}_{i=1}^N$.

Note that certain other generative models, such as Variational Auto-Encoder (VAE) [45], Generative Adversarial Networks (GAN) [46, 47], and Denoising Diffusion Probabilistic Models (DDPM) [48, 49], may be better at generating data. We introduce the model-based offline RL primarily because it has been shown to generate violating data with exploration [38] and possess the ability to process time-series features efficiently.

### 4.3 Safe-Critical Decision Making with Constraints.

To train offline CT, we gather the medical expert dataset $\mathcal{D}_e$ from the environment. Then, we employ gradient descent to train the model-based offline RL, guided by Equation 7 and Equation 8, continuing until the model converges. Using this RL model, we automatically generate violating data denoted as $\mathcal{D}_v$. Subsequently, CT is optimized based on Equation 6 to get the cost function $C$, leveraging samples from both $\mathcal{D}_e$ and $\mathcal{D}_v$. To learn a safe policy, we train the policy $\pi$ using $C$ until it converges based on Equation 1. The detailed training procedure is presented in Algorithm 1.

## 5 Experiment

In this section, we first provide a brief overview of the task, as well as data extraction and preprocessing. Subsequently, in Section 5.1, we demonstrate that CT can describe constraints in healthcare and capture critical patient states. We emphasize its applicability to various CRL methods and its ability to approach the optimal policy for reducing mortality rates in Section 5.2. Finally, Section 5.3 discusses the realization of the objective of safe medical policies.

**Algorithm 1** Safe Policy Learning with Offline CT
___
**Input:** Expert trajectories $\mathcal{D}_e$, context length $K$, target reward $\hat{R}_1$, samples $N$, episode length $T$
1: Train model-based offline RL $\mathcal{M}$: Update $\vartheta$, $\varphi$ and $\mu$ using the Equation (7) and Equation (8)
2: **for** t = 1,...,T **do**
3:    Sample initial states $S_1$ from $\mathcal{D}_e$
4:    Generate the violating dataset: $\mathcal{D}_v \leftarrow \mathcal{M}.\text{generate\_data}(S_1, \hat{R}_1, K)$
5:    Sample set of trajectories $\{\tau_e^{(i)}\}_{i=1}^N$ and $\{\tau_v^{(i)}\}_{i=1}^N$ from $\mathcal{D}_e$ and $\mathcal{D}_v$
6:    Train offline CT: Use $\{\tau_e^{(i)}\}_{i=1}^N$ and $\{\tau_v^{(i)}\}_{i=1}^N$ to update $\phi$ based on Equation (6)
7:    Safe policy learning: Update $\pi$ using the cost function $C_\phi(\tau)$ based on Equation (1)
8: **end for**
**Output:** $\pi$ and $C(\tau)$
___

**Tasks.** We primarily use the sepsis task that is commonly used in previous works [9, 20, 42, 22], and supplement some experiments on the mechanical ventilator task [23, 50]. The detailed definition of the two tasks mentioned above can be found in Appendix B.1 and B.2.

**Data Extraction and Pre-processing.** Our medical dataset is derived from the Medical Information Mart for Intensive Care III (MIMIC-III) database [51]. For each patient, we gather relevant physiological parameters, including demographics, lab values, vital signs, and intake/output events. Data is grouped into 4-hour windows, with each window representing a time step. In cases of multiple data points within a step, we record either the average or the sum. We eliminate variables with significant missing values and use the $k$-nearest neighbors method to fill in the rest. Notably, the training dataset consists of data from surviving patients, while the validation set includes survivors and non-survivors.

**Model-based Offline RL Evaluation.** To ensure the rigor of the experiments, we evaluate the validity of the model-based offline RL, as detailed in Appendix B.3.

### 5.1 Can Offline CT Learn Effective Constraints?

In this section, we primarily assess the efficacy of the cost function learned by offline CT in sepsis, focusing particularly on its capability to evaluate patient mortality rates and capture critical events. First, we employ the cost function to compute cost values for the validation dataset. Subsequently, we statistically analyze the relationship between these cost values and mortality rates. As shown in Figure 5, there is an increase in patient mortality rates with rising cost values. It's noteworthy that such increases in mortality rates are often attributed to suboptimal medical decisions. Therefore, these experimental findings affirm that the cost values effectively reflect

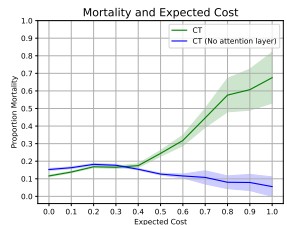
Figure 5: The relationship between cost and mortality.

the quality of medical decision-making. To observe the impact of the attention layer (non-Markovian layer), we conduct experiments by removing the attention layer from CT. The results reveal that the penalty values do not correlate proportionally with mortality rates. This indicates that the attention layer plays a crucial role in assessing constraints.

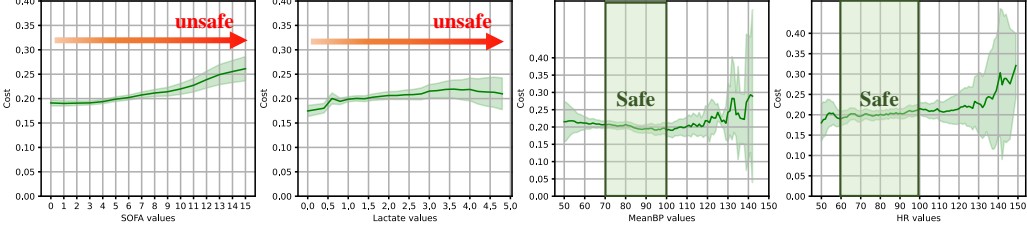

Figure 6: The relationship between physiological indicators and cost values. As SOFA and lactate levels become increasingly unsafe, the cost increases. Mean BP and HR at lower values within the safe range incur a lower cost, but as they move into unsafe ranges, the cost increases, penalizing previous state-action pairs. The cost can differentiate between relatively safe and unsafe regions.

To assess the capability of the cost function to capture key events, we analyze the relationship between physiological indicators and cost values. We focus on four key indicators in sepsis treatment: Sequential Organ Failure Assessment (SOFA) score [52], lactate levels [53], Mean Arterial Pressure

(MeanBP) [54], and Heart Rate (HR) [55]. The SOFA score and lactate levels are critical indicators for assessing sepsis severity, with higher values indicating greater patient risk. MeanBP and HR are essential physiological metrics, typically ranging from 70 to 100 mmHg and 60 to 100 beats, respectively. Deviations from these ranges can signify patient risk. As depicted in Figure 6, the cost values effectively distinguish between high-risk and safe conditions, reflecting changes in patient status. Additional details on other parameters' relationship with cost are in Appendix B.4.

### 5.2 Can Offline CT Improve the Performance of CRL?

**Baselines.** We adopt the DDPG method as the baseline in sepsis research [9], and the Double Deep Q-Learning (DDQN) and Conservative Q-Learning (CQL) methods as baselines in ventilator research [23]. Since there are no other offline inverse reinforcement learning works available for reference, we have included two additional settings: no cost and custom cost. In the case of no cost, the cost is set to zero, while the design of custom constraints is outlined in Appendix A. These settings help evaluate whether CT can infer effective constraints.

**Metrics.** To assess effectiveness, we use $\omega$ to indicate the probability that the policy is optimal and analyze the relationship between DIFF and mortality rate through a graph. Recently, Kondrup *et al.* [23] use the Fitted Q Evaluation (FQE) [56] to evaluate the policy in healthcare. However, the value estimates of FQE depend solely on the dataset $\mathcal{D}$ and the actions chosen by the policy $\pi$ used to train FQE. This reliance can lead to inaccurate estimates when evaluating unseen state-action pairs. Therefore, we do not adopt this method as an evaluation metric.

**Results.** We combine our method CT with common CRL algorithms (e.g., VOCE, COpiDICE, BCQ-Lag, and CDT), and compare them with both no-cost and custom cost settings. Each CRL model is trained using no cost, custom cost, and CT separately, with other parameters set the same during training. For evaluation metrics, we use IV difference (IV DIFF), vaso difference (VASO DIFF), and combined [IV, VASO] difference (ACTION DIFF) as the metrics to be ranked. We measure the mean and variance of $\omega\%$ in 10 sets of random seeds, and the results are shown in Table 2. From the results, we can conclude: (1) In different CRL methods, CT consistently makes the strategy closer to the one with lower mortality rates, with a probability $8.85\%$ higher than DDPG. (2) We find that CDT+CT achieves better results on all three metrics. CDT is also a transformer-based method, which indicates that transformer-based architecture indeed exhibits more outstanding performance in healthcare.

Figure 7 illustrates the relationship between IV and VASO DIFF with mortality rates under the DDPG and CDT+CT methods in sepsis. In VASO DIFF, when the gap is zero, the mortality rate under CDT+CT is lower than that under DDPG, indicating that following the former strategy could lead to a lower mortality rate. Similarly, in IV DIFF, the same trend is observed. Notably, for the IV strategy, the lowest mortality rate for DDPG does not occur at the point where the difference is zero, indicating a significant estimation bias.

Table 2: Performance of sepsis strategies under various offline CRL models and different constraints.

| $\omega\%$ | COST | IV DIFF ↑ | VASO DIFF ↑ | ACTION DIFF ↑ |
|---|---|---|---|---|
| DDPG | - | 50.95±1.34 | 51.45±0.75 | 51.15±1.15 |
| VOCE | No cost | 47.45±0.52 | 46.35±1.82 | 51.00±0.86 |
| | Custom cost | 46.45±0.46 | 52.00±0.98 | 49.40±1.04 |
| | CT | **53.33±0.94** | **59.04±1.13** | **56.15±1.08** |
| CopiDICE | No cost | 48.30±0.91 | 60.10±0.6 | 51.25±0.70 |
| | Custom cost | **53.05±1.35** | 55.20±0.24 | 53.90±1.04 |
| | CT | 51.95±0.41 | **60.85±1.08** | **54.60±0.60** |
| BCQ-Lag | No cost | 47.50±1.32 | 51.05±0.61 | 49.35±1.08 |
| | Custom cost | 51.54±0.16 | **56.23±1.43** | 53.69±1.62 |
| | CT | **52.45±1.01** | 55.34±1.20 | **54.39±0.86** |
| CDT | No cost | 56.50±0.81 | 62.45±1.20 | 58.90±1.34 |
| | Custom cost | 54.70±1.12 | 59.85±1.51 | 57.80±1.00 |
| | CT | **57.15±1.67** | **65.20±1.22** | **60.00±1.49** |
| CDT | Without CT | 56.50±0.81 | 62.45±1.20 | 58.90±1.34 |
| CDT | No attention layer | 55.25±1.46 | 64.00±1.54 | 57.90±0.78 |
| Generative Model | - | 55.49±2.55 | 56.60±1.33 | 57.00±2.06 |

**Blue:** Safe policy is closer to the optimal policy. ↑: higher is better.

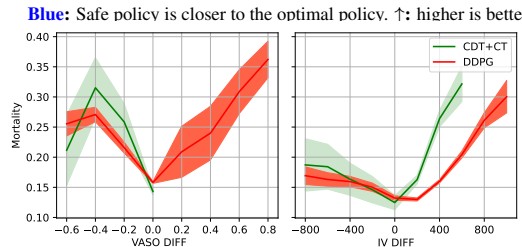

Figure 7: The relationship between DIFF and the mortality rate in sepsis. The x-axis represents the DIFF. The y-axis indicates the mortality rate of patients at a given DIFF. The solid line represents the mean, while the shaded area indicates the Standard Error of the Mean (SEM).

In addition, corresponding experiments are conducted on the mechanical ventilator, as shown in Figure 8. Compared to previous methods DDQN and CQL, under the CDT+CT approach, a noticeable trend is observed where the proportion of mortality rates increases with increasing differences. When

there is a significant difference in DIFF, the results may be unreliable, possibly due to the limited data distribution in the tail.

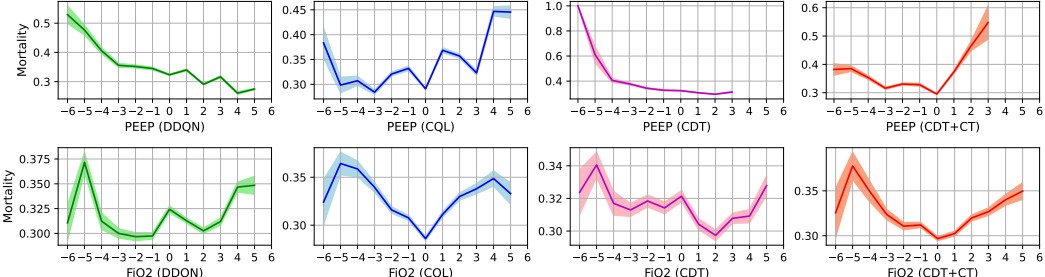

Figure 8: The relationship between the DIFF of actions and mortality in mechanical ventilator. The actions mainly consist of Positive End Expiratory Pressure (PEEP) and Fraction of Inspired Oxygen (FiO2), which are crucial parameters in ventilator settings.

### 5.3 Can CRL with Offline CT Learn Safe Policies?

We have confirmed the existence of two unsafe strategy issues, namely "too high" and "sudden change" in the treatment of sepsis, particularly in vaso in Section 1. To validate whether the CRL+CT approach could address these concerns, we employ the same statistical methods to

Table 3: The proportion of "too high" and "sudden change" occurrences in drug dosage recommended by RL methods.

| Drug dosage $(\mu g/(kg \cdot min))$ | Physician | DDPG | CDT No cost | CDT Custom cost | CT |
|---|---|---|---|---|---|
| vaso > 0.75 | 2.27% | 7.44% | 0.13% | 0% ↓ | 0% ↓ |
| vaso > 0.9 | 1.71% | 7.40% | 0.09% | (max = 0.00) | (max = 0.11) |
| Δ vaso > 0.75 | 2.45% | 21.00% | 0.64% | 0% ↓ | 0% ↓ |
| Δ vaso > 0.9 | 1.88% | 20.62% | 0.48% | (max Δ = 0.00) | (max Δ = 0.10) |

evaluate our methodology, shown in Table 3. To elucidate the efficacy of CT, we compare it with CDT+No-cost and CDT+Custom-cost approaches. We find that only the custom cost and CT methods successfully mitigated the risks associated with "too high" and "sudden change" behaviors. However, the custom cost approach opts to avoid administering drugs to mitigate these risks. Without these drugs, the patient's condition may not be alleviated, potentially leading to patient mortality. The CDT+CT approach can give a more appropriate drug dosage.

**Ablation Study.** To investigate the impact of each component on the model's performance, we conducted experiments by sequentially removing each component from the CDT+CT model. The results are presented in the lower half of Table 2. Both CT and its non-Markovian layer (attention layer) are indispensable and crucial components; removing either one results in a decrease in performance. Additionally, we observed that even a pure generative model outperforms DDPG in terms of performance. This is primarily because it inherently operates as a sequence-based reinforcement learning model, possessing exploration and consideration for long-term history. Therefore, this further underscores the effectiveness of sequence-based approaches in healthcare applications.

## 6 Conclusion

In this paper, we propose offline CT, a novel ICRL algorithm designed to address safety issues in healthcare. This method utilizes a causal attention mechanism to observe patients' historical information, similar to the approach taken by actual doctors and employs non-Markovian importance weights to effectively capture critical states. To achieve offline learning, we introduce a model-based offline RL for exploratory data augmentation to discover unsafe decisions and train CT. Experiments in sepsis and mechanical ventilation demonstrate that our method avoids risky behaviors while achieving strategies that closely approximate the lowest mortality rates.

**Limitations.** There are also several limitations of offline CT: (1) Lack of rigorous theoretical analysis: We did not precisely define the types of constraint sets, thereby conducting rigorous theoretical analysis on constraint sets remains challenging; (2) Need for more computational resources: Due to the Transformer architecture, more computational resources are required; (3) Fewer evaluation metrics: There is a lack of more medical-specific evaluation metrics in the experimental evaluation section; (4) Unrealistic assumptions of expert demonstrations: we assume that expert demonstrations are optimal in both constraint satisfaction and reward maximization. However, in reality, this assumption may not always hold. Therefore, researching a more effective approach to address the aforementioned issues holds promise for the field of secure medical reinforcement learning.

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

## A    Design and Analysis of the Custom Constraint Function

We base our design on prior knowledge that intravenous (IV) intake exceeding $2000mL/4h$ or vasopressor (Vaso) dosage surpassing $1g/(kg \cdot min)$ is generally considered unsafe in sepsis treatment [6]. To design a reasonable constraint function, we refer to the constraint function designed by Liu *et al.* in the Bullet safety gym environments[38]. We define the cost function as shown in Equation 9. Thus, during the treatment of sepsis, if the agent exceeds the maximum dosage thresholds of the two medications, it incurs a cost due to constraint violation.

$$c\left(s,a\right) = \mathbf{1}\left(a_{IV} > a_{IV \ \max}\right) + \mathbf{1}\left(a_{Vaso} > a_{Vaso \ \max}\right) \tag{9}$$

where, $s$ and $a$ represent the patient's state and action, respectively. $a_{IV \ \max} = 2000$ indicates that the maximum fluid intake through IV is $2000mL$, and $a_{Vaso \ \max} = 1$ signifies that the maximum Vaso dosage is $1\mu g/(kg \cdot min)$.

We applied our custom constraint function in the CDT [38] method, and the results are shown in Figure 9. Compared to the Vaso dosage recommended by doctors, our strategy exhibits excessive suppression of the Vaso. The maximum dosage of Vaso is $0.0011\mu g/(kg \cdot min)$, which is minimal and insufficient to provide the patient with effective therapeutic effects.

Therefore, Equation 9 is not suitable. The primary issues may include uniform constraint strength for excessive drug dosages, for instance, the cost for IV exceeding 2000 mL and IV exceeding 3000 mL is the same at 1; lack of generalization, where the constraint cost does not vary with the patient's tolerance. If a patient has an intolerance to VASO, the maximum value for VASO maybe 0, which cannot be captured by the self-imposed constraint function. Moreover, it lacks generalization, requiring redesign of the constraint function when addressing other unsafe medical issues; and it's essential to ensure the correctness of the underlying medical knowledge premises.

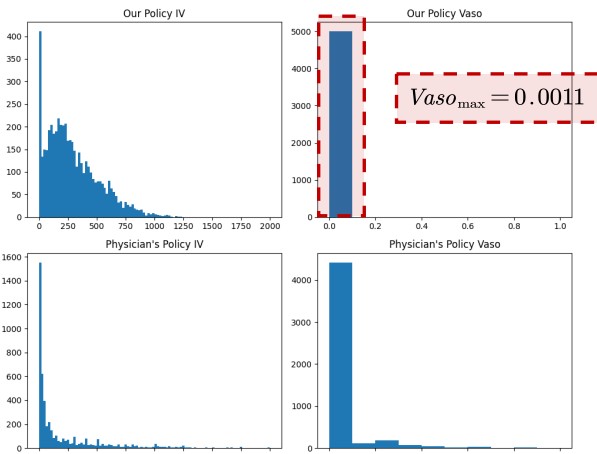

Figure 9: Drug dosage distribution under custom constraint functions in sepsis.

## B    Experiment Supplement

### B.1    Sepsis Problem Define

Our definition is similar to [40]. We extract data from adult patients meeting the criteria for sepsis-3 criteria [57] and collect their data within the first 72 hours of admission.

**State Space.** We use a 4-hour window and select 48 patient indicators as the state for a one-time unit of the patient. The state indicators include Demographics/Static, Lab Values, Vital Signs, and Intake and Output Events, detailed as follows [40]:

- Demographics/Static: Shock Index, Elixhauser, SIRS, Gender, Re-admission, GCS - Glasgow Coma Scale, SOFA - Sequential Organ Failure Assessment, Age

- Lab Values Albumin: Arterial pH, Calcium, Glucose, Hemoglobin, Magnesium, PTT - Partial Thromboplastin Time, Potassium, SGPT - Serum Glutamic-Pyruvic Transaminase, Arterial Blood Gas, BUN Blood Urea Nitrogen, Chloride, Bicarbonate, INR - International Normalized Ratio, Sodium, Arterial Lactate, CO2, Creatinine, Ionised Calcium, PT - Prothrombin Time, Platelets Count, SGOT Serum Glutamic-Oxaloacetic Transaminase, Total bilirubin, White Blood Cell Count

- Vital Signs: Diastolic Blood Pressure, Systolic Blood Pressure, Mean Blood Pressure, PaCO2, PaO2, FiO2, PaO/FiO2 ratio, Respiratory Rate, Temperature (Celsius), Weight (kg), Heart Rate, SpO2

- Intake and Output Events: Fluid Output - 4 hourly period, Total Fluid Output, Mechanical Ventilation

**Action Space.** Regarding the treatment of sepsis, there are two main types of medications: intravenous fluids and vasopressors. We select the total amount of intravenous fluids for each time unit and the maximum dose of vasopressors as the two dimensions of the action space, defined as $(\text{sum(IV)}, \max(\text{Vaso}))$. Each dimension is a continuous value greater than $0$.

**Reward Function.** We refer to the reward function used in [9], as shown in the following equation:

$$r\left(s_t, s_{t+1}\right) = \lambda_1 \tanh\left(s_t^{\text{SOFA}} - 6\right) + \lambda_2 \left(s_{t+1}^{\text{SOFA}} - s_t^{\text{SOFA}}\right)) \tag{10}$$

Where $\lambda_0$ and $\lambda_1$ are hyperparameters set to $-0.25$ and $-0.2$, respectively. This reward function is designed based on the SOFA score, as it is a key indicator of the health status for sepsis patients and widely used in clinical settings. The formula describes a penalty when the SOFA score increases and a reward when the SOFA score decreases. We set $6$ as the cutoff value because the mortality rate sharply increases when the SOFA score exceeds 6 [58].

## B.2 Mechanical Ventilation Treatment Problem Define

The RL problem definition for Mechanical Ventilation Treatment is referenced from [23].

**State Space.**

- Demographics/Static: Elixhauser, SIRS, Gender, Re-admission, GCS, SOFA, Age

- Lab Values Albumin: Arterial pH, Glucose, Hemoglobin, Magnesium, PTT, BUN Blood Urea Nitrogen, Chloride, Bicarbonate, INR, Sodium, Arterial Lactate, CO2, Creatinine, Ionised Calcium, PT, Platelets Count, White Blood Cell Count, Hb

- Vital Signs: Diastolic Blood Pressure, Systolic Blood Pressure, Mean Blood Pressure, Temperature, Weight (kg), Heart Rate, SpO2

- Intake and Output Events: Urine output, vasopressors, intravenous fluids, cumulative fluid balance

**Action Space.** The action space mainly consists of Positive End Expiratory Pressure (PEEP) and Fraction of Inspired Oxygen (FiO2), which are crucial parameters in ventilator settings. Here, we consider a discrete space configuration, with each parameter divided into 7 intervals. Therefore, our action space is $7 \times 7$, depicted as 4.

Table 4: The action space of the mechanical ventilator.

| Action | 0 | 1 | 2 | 3 | 4 | 5 | 6 |
|---|---|---|---|---|---|---|---|
| PEEP(cmH20) | 0-5 | 5-7 | 7-9 | 9-11 | 11-13 | 13-15 | >15 |
| FiO2(Percentage(%)) | 25-30 | 30-35 | 35-40 | 40-45 | 45-50 | 50-55 | >55 |

**Reward Function.** The primary objective of setting respiratory parameters is to ensure the patient's survival. We adopt the same reward function design as the work [23], defined as Equation 11. This reward function first considers the terminal reward: if the patient dies, the reward $r$ is set to $-1$; otherwise, it is $+1$ in the terminal state. Additionally, to provide more frequent rewards, intermediate rewards are considered. Intermediate rewards mainly focus on the Apache II score, which evaluates

various parameters to describe the patient's health status. This reward function utilizes the increase or decrease in this score to reward the agent.

$$r\left(s_t, a_t, s_{t+1}\right) = \begin{cases} +1 & \text{if } t = T \text{ and } m_t = 1 \\ -1 & \text{if } t = T \text{ and } m_t = 0 \\ \frac{(A_{t+1} - A_t)}{\max_A - \min_A} & \text{otherwise} \end{cases} \tag{11}$$

In Equation 11, $T$ represents the length of the patient's trajectory, $m$ indicates whether the patient ultimately dies, $A$ denotes the Apache II score, and $\max_A$ and $\min_A$ respectively denote the maximum and minimum values.

### B.3 The Evaluation of Model-based Offline RL

**Generating data within a reasonable range.** To validate model-based offline RL, we first check whether the values it produces fall within the legal range. The results are depicted in Figure 10. After analyzing the generated data, we find that the majority of state values have a probability of over 99% of being within the legal range. A few values related to gender and re-admission range between 60% and 70%. This could be due to these two indicators having limited correlation with other metrics, making them more challenging for the model to assess.

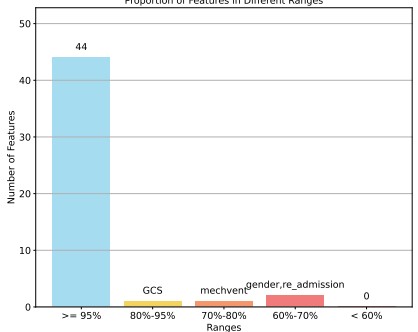

Figure 10: The accuracy of predicting different state values within the legal range.

Figure 11: The relationship between average prediction error and trajectory length.

**Generating violating data.** In addition, we evaluate the violating actions generated by the model, as shown in Figure 12. When compared with expert strategies and penalty distributions, we find that the actions generated by the model mostly fall within the legal range. However, it occasionally produces behaviors that are inappropriate for the current state, constituting violating data. This indicates that our generative model can produce legally violating data.

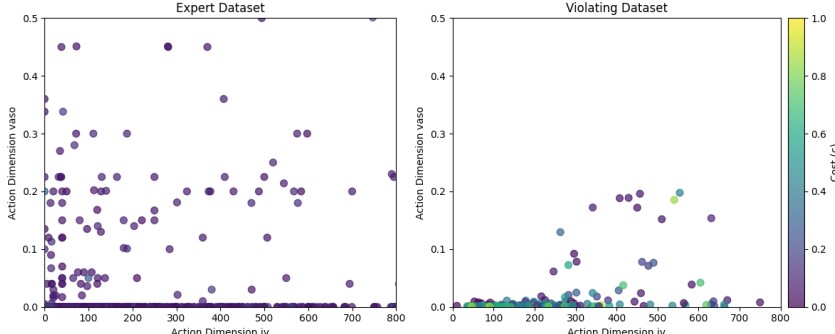

Figure 12: The distribution and penalty values of violating data and expert data.

**The length of a trajectory.** Regarding the selection of trajectory length, we consider the relationship between the average prediction error, the error of the last point in the trajectory, and the trajectory

length. We use the model-based offline RL to generate trajectories and compare them with expert data using the Euclidean distance to measure their differences. We evaluate the average error and the error of the last point in the trajectory, as shown in Figure 11. We observe that with an increase in trajectory length, the average prediction error at each time step decreases, while the state error stabilizes. Taking into account the observation length and prediction accuracy, we ultimately choose to generate trajectories with lengths ranging from 10 to 15.

## B.4 The Evaluation of Cost function in Sepsis

To validate that the CT method captures key states, we conduct statistical analysis on the relationship between state values and penalty values. We collect penalty values under different state values for all patients, and the complete information is shown in Figure 13. We find that the CT method successfully captures unsafe states and imposes higher penalties accordingly. The safe range of state values is shown in Table 5.

To validate the role of the attention layer in capturing states in CT, we conducted tests, and the experimental results are presented in Figure 14 and 13. We found that the attention layer plays a crucial role in state capture. For instance, in the case of an increase in the SOFA score, without the attention layer, this increase cannot be captured, while with the attention layer, it clearly captures the change. Thus, this indicates that SOFA, as a key diagnostic indicator of sepsis, with the help of the attention layer, CT can accurately capture its changes.

Table 5: State indicators and their normal ranges.

| Indicator | Safe Range | Indicator | Safe Range | Indicator | Safe Range |
|---|---|---|---|---|---|
| Albumin | 3.5~5.1 | HCO3 | 25~40 | SGOT | 0~40 |
| Arterial_BE | -3~+3 | Glucose | 70~140 | SGPT | 0~40 |
| Arterial_lactate | 0.5~1.7 | HR | 60~100 | SIRS | ↓ |
| Arterial_PH | 7.35~7.45 | Hb | 12~16 | SOFA | ↓ |
| BUN | 7~22 | INR | 0.8~1.5 | Shock_Index | ↓ |
| CO2_mEqL | 20~34 | MeanBP | 70~100 | Sodium | 135~145 |
| Calcium | 8.6~10.6 | PT | 11~13 | SpO2 | 95~99 |
| Chloride | 96~106 | PTT | 23~37 | SysBP | 90~139 |
| Creatinine | 0.5~1.5 | PaO2_FiO2 | 400~500 | Temp_C | 36.0~37.0 |
| DiaBP | 60~89 | Platelets_count | 125~350 | WBC_count | 4~10 |
| FiO2 | 0.5~0.6 | Potassium | 4.1~5.6 | PaCO2 | 35~45 |
| GCS | ↑ | RR | 12~20 | PaO2 | 80~100 |

↑ indicates higher values are more normal, while ↓ indicates lower values are more normal.
The maximum value for GCS is 15. The minimum value for SIRS, SOFA, and Shock_Index is 0.

## B.5 Experimental Settings

To train the CRL+CT model, we use a total of 3 NVIDIA GeForce RTX 3090 GPUs, each with 24GB of memory. Training a CRL+CT model typically takes 5-6 hours. We employ 5 random seeds for validation. We use the Adam optimization algorithm to optimize all our networks, updating the learning rate using a decay factor parameterization at each iteration. The main hyperparameters are summarized in Table 6 and 7.

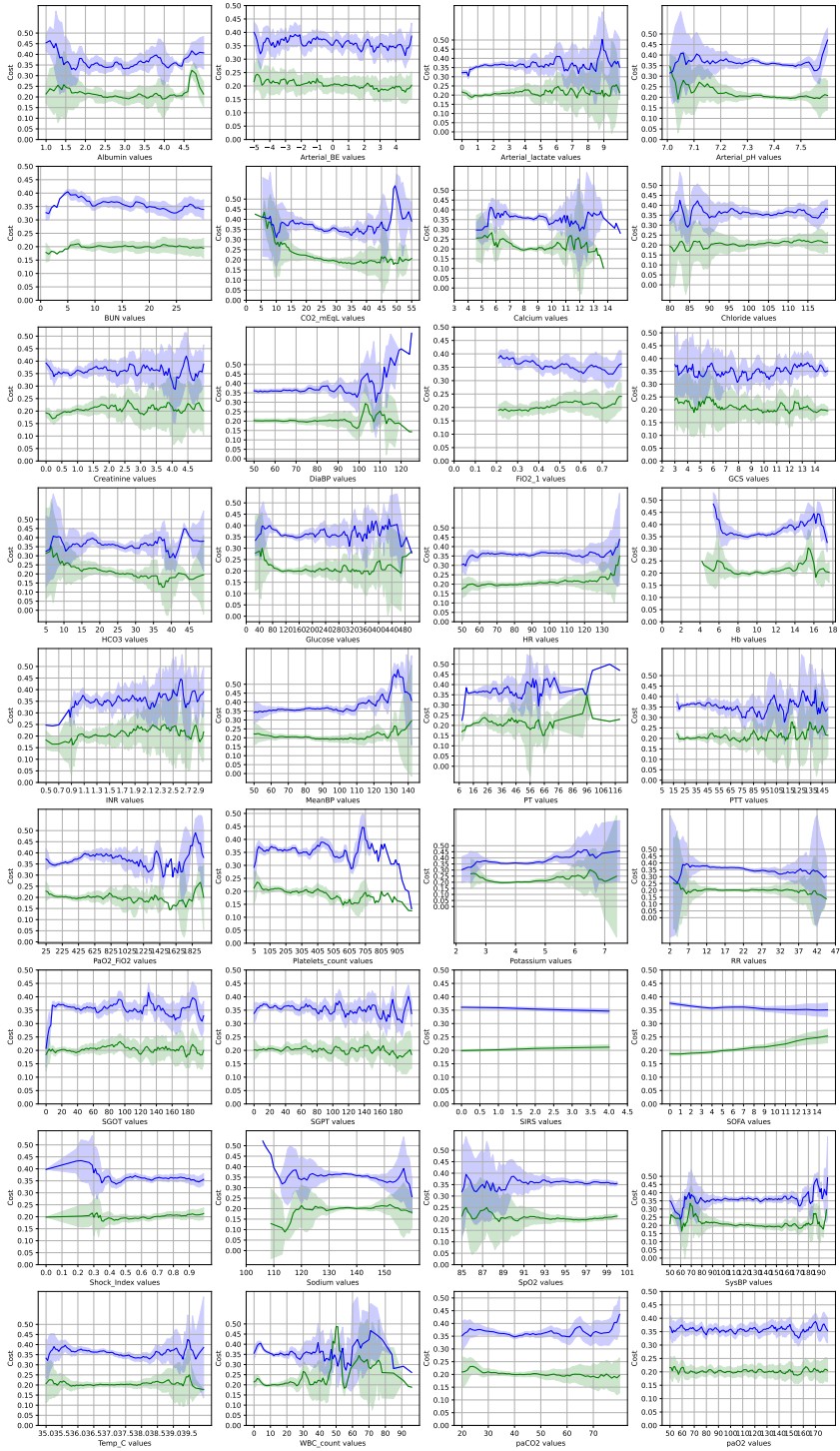

Figure 13: The relationship between all states and cost values

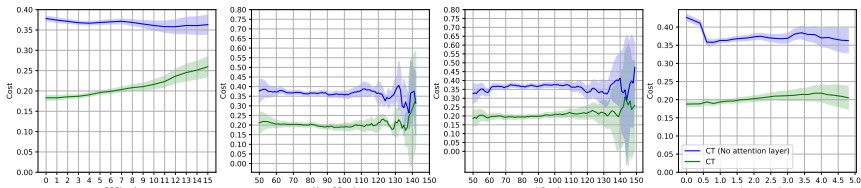

Figure 14: The performance contrast between CT with and without an attention layer. The blue line represents the absence of an attention layer, while the green line indicates the presence of an attention layer.

Table 6: List of the utilized hyperparameters in CT.

| Offline CT Parameters | values |
|---|---|
| Genetivate Model | |
| Embedding_dim | 128 |
| Layer | 3 |
| Head | 8 |
| Learning rate | 1e-4 |
| Pre-train steps | 5000 |
| Batch size | 256 |
| CT | |
| Embedding_dim | 64 |
| Layer | 3 |
| Head | 1 |
| Learning rate | 1e-6 |
| Update steps | 30000 |
| Batch size | 512 |
| CDT | |
| Learning rate | 1e-4 |
| Embedding_dim | 128 |
| Layers | 3 |
| Heads | 8 |
| Update steps | 60000 |

Table 7: List of the utilized hyperparameters in CRL.

| Parameters | Sepsis | Parameters | Mechanical Ventilation |
|---|---|---|---|
| General | | General | |
| Expert data patient number | 14313 | Expert data patient number | 13846 |
| Validation data patient number | 6275 | Validation data patient number | 5954 |
| Max Length | 10 | Max Length | 10 |
| Action_dim | 2 | Action_dim | 2 |
| State_dim | 48 | State_dim | 36 |
| Gamma | 0.99 | Gamma | 0.99 |
| DDPG | | DDQN | |
| Learning rate | 1e-3 | Learning rate | 1e-4 |
| Policy Network | 256,256 | Policy Network | 64,64 |
| Replay memory size | 20000 | Update steps | 500000 |
| Update steps | 20000 | | |
| VOCE | | CQL | |
| Learning rate | 1e-3 | Learning rate | 1e-4 |
| Policy Network | 256,256 | Policy Network | 64,64 |
| Alpha scale | 10 | Update steps | 500000 |
| KL constraint | 0.01 | Alphas | 0.05,0.1,0.5,1,2 |
| Dual constraint | 0.1 | | |
| Update steps | 4000 | | |
| CopiDICE | | | |
| Learning rate | 1e-4 | | |
| Policy Network | 256,256 | | |
| Alpha | 0.5 | | |
| Cost limit | 10 | | |
| Update steps | 100000 | | |
| BCQ-Lag | | | |
| Learning rate | 1e-3 | | |
| Policy Network | 256,256 | | |
| Cost limit | 10 | | |
| Lambda | 0.75 | | |
| Beta | 0.5 | | |
| Update steps | 100000 | | |

