# OpenReview forum: "Offline Inverse Constrained Reinforcement Learning for Safe-Critical Decision Making in Healthcare"
_NeurIPS.cc/2024/Conference — Submitted to NeurIPS 2024_

### Official Review · Reviewer_cbZJ · 2024-07-12

**Soundness:** 1
**Presentation:** 1
**Contribution:** 2
**Rating:** 3
**Confidence:** 5

**Summary:**

This paper studies inverse RL with learnable constraints in the offline setting, focusing on practical applications in healthcare tasks. The main approach appears to be combining the decision transformer architecture in the offline RL literature with inverse constrained RL with max entropy framework. Experiments were conducted on two healthcare tasks: sepsis and mechanical ventilation.

**Strengths:**

Studies an interesting and important problem.

**Weaknesses:**

There is significant issues with the writing: the text is incoherent throughout and really hindered my understanding of the paper.

**Questions:**

1. Unclear what exactly is the "gap" the paper addresses. It mentions several issues including Markov assumption and history dependence, personalization & individual differences, offline setting, decision transformers, and it's unclear what the focus is. Otherwise, the paper seems to be applying decision transformer architecture to the constrained IRL problem.

1. Table 1: "too high" associated with increased mortality making it an "unsafe behavior" - this seems like confounding, patients are sick and that's why they receive high dose. Calling it "unsafe behavior" without conditioning on the patient state is inappropriate.

2. Claims about Markovianity: if Markov assumption does not hold, it seems to suggest the state is not capturing enough information about the "environment state".
    - L49: "The Markov decision is not compatible with medical decisions." and "historical states of patients are crucial for medical decision" why can't one incorporate history into states?
    - Fig 1: if within the same state different treatments should be used due to individual differences, that also suggests the state definition is not capturing individual differences (L57 "ignore individual differences") and you should redefine the states.
3. Organization & Flow:
    - page 4 Sec 4 Methods talks about ICRL - is this your proposed method or previous work? If it's previous work it should be stated clearly and probably does not belong in Methods.
    - L116 - when talking about Sec 3 Problem Formulation of constrained MDP, the text talks about "we extract data within 72 hours of patient admission, with each 4-hour interval" - this seems like experimental details rather than the mathematical setup of your method.
4. Writing and notation: writing is incoherent, many notations are not defined.
    - L122 cost c_t \sim C, shouldn't cost depend on (s,a)?
    - Eqn (2) in the definition of "probability of approaching the optimal policy", what does "top N" mean? Top N according to what? What is N? If N is dataset size, how do you "collect 2N from offline dataset" (L144)?
    - L145 "calculate the DIFF and sort it in ascending order" Is the DIFF just one number? How do you sort it? What are you sorting?
    - Eqn (3) what is R(tau), L157 is it ZM or ZMC, what is β
    - L162 "where M ζθ denotes the MDP obtained after augmenting M with the cost function Cθ" Do you mean ζθ is the same as Cθ? What is π_Mζθ (L163)?
    - L202 "doesn't" contractions should be avoided in academic writing
    - L206 "This layer is employed to define the cost weight for non-Markovians" What is "non-Markovians"?
    - L207 What is "causality transformer"? In "generate the importance weights", importance weights has a specific meaning in RL. The authors probably meant the cost weights.
    - L219 "We construct an expert and a violating dataset to evaluate Equation 6 in offline." This sentence doesn't make sense grammatically.
    - L248 "generating violating data" have you defined violating data? "may incentivize": why does the procedure incentivize generation of "violating data", do you show it theoretically or empirically?
    - L283 why "Notably, the training dataset consists of data from surviving patients" Please provide more justification for this. Training on only surviving patients would limit the dataset and state-actions available to the learning agent.
5. Questions about experiments:
    - L297 if cost values are positively correlated with mortality rates which is the reward, what's the point of having a separate cost function instead of just using the reward?
    - L300 "This indicates that the attention layer plays a crucial role in assessing constraints." could you elaborate why?
    - L321 "FQE cannot evaluate unseen (s,a)" are you saying your approach can evaluate unseen (s,a)?
    - L331 "combined DIFF": vaso and IV have very different scales, how do you combine?
    - L357 "corresponding experiments are conducted on the mechanical ventilator". What?
    - It's unclear what Fig 8 is showing. Table 3 not explained, unclear if lower or higher is better, what is max ∆?

**Limitations:**

Limitations are discussed.

---

> ### Author Rebuttal · Authors · 2024-08-06
>
> **Response 1:** We did indeed introduce our challenge by raising several issues, where the challenges are the key issue we aim to address. To aid your understanding, we will briefly re-explain it as follows: Current RL methods display risky behavior, which we aim to mitigate using Constrained RL (CRL). Effective CRL implementation depends on acquiring constraints, but existing custom constraint functions lack personalization, making ICRL a promising solution. However, it encounters difficulties in medical scenarios due to the Markov assumption and history dependence, limiting its effectiveness in healthcare.
>
> **Response 2:** By "too high" we meant a drug dosage that is dangerous for any patient state. We fully agree with the reviewer's comment that "unsafe behavior without conditioning on the patient state is inappropriate." Therefore, we have taken this into careful consideration in our design and have developed a personalized cost function $C(\tau)$, which assigns a cost to the current drug dosage based on the patient's historical treatment trajectory.
>
> **Response 3:** Incorporating history into the state or redefining the state are the most direct approaches. However, there may be the following issues: (1) Incorporating history into the state can lead to redundant calculations and increased complexity. For a patient's trajectory, the state at timestamp t will include the previous t-1 timestamps, which can result in excessive redundancy. The number of timestamps stored increased from O(n) to O(n^2). (2) Latent states might be unobservable. As Reviewer 1 pointed out, the patient system cannot be modeled based on a Markov process due to the high-dimensional latent states that cannot be represented.
>
> **Response 4:** (1) Thank you for your suggestion. ICRL is part of previous work, which we have mentioned in the Introduction. We will reiterate here. (2) This is indeed a description of the experimental setup.
>
> **Response 5:**
>
> **1. L22** Not necessarily; it could be either $(s,a)$ or trajectory $\tau$. Therefore, we do not impose any restrictions here.
>
> **2. L45** It is explained that 2N represents selecting 2N patients from the dataset, therefore N represents a constant, and 2N is less than or equal to the size of the dataset. And among these 2N patients, N patients died under the doctor's treatment, and N patients survived. We calculate the DIFF for the 2N patients and rank them in ascending order of DIFF. The top N patients are then selected as the top N.
>
> **3. L162** We will restate $\omega$ to solve your questions: (1) Select 2N patients from the dataset. And among these 2N patients, N patients died under the doctor's treatment, and N patients survived. (2) In the dataset, we know the patient's state and the drug dosage (a) under the doctor's policy. We use the estimated policy to provide the drug dosage (b) for the same patient state. (3) Calculate the DIFF for each patient, defined as $DIFF = b−a$. For the 2N patients, we can obtain 2N DIFF values. (4) Sort the 2N DIFF values in ascending order and observe the survival status of the top N patients recorded in the dataset. (5) The top N patients indicate that the difference between our policy and the doctor's policy is small. And if the survival rate of these top N patients is higher, it suggests that our policy is closer to the ideal optimal policy. If our policy is ideal, the top N patients, once sorted, will have the highest survival rate. (6) In addition, we also need to consider the size of the DIFF values. For surviving patients, a smaller DIFF difference is preferable.
>
> **4. L57** (1) $R(\tau)$ is the reward of the trajectory $\tau$. (2) L157 is $Z_{\mathcal{M}^{\mathcal{C}}}$. (3) $\beta$ is a parameter describing how close the agent is to the optimal distribution.
>
> **5. L162** The cost function can be formulated as $C_{\theta}=1–\zeta_{\theta}$. $\mathcal{M}^{\zeta_{\theta}}$ is the MDP that results from adding the cost function $C_{\theta}$ to the original MDP $\mathcal{M}$. The executing policy for this augmented MDP is denoted as $\pi_{\mathcal{M}^{\hat{\zeta}_{\theta}}}$.
>
> **6. L202** We will make corrections.
>
> **7. L206** "Non-Markovians" refers to scenarios where the future state depends on both current and past states or actions, unlike "Markovian" processes, which depend solely on the current state.
>
> **8.** (1) "causality transformer" is the casual transformer, which refers to a variant of the Transformer model that incorporates causal (or temporal) relationships into its architecture. (2) Yes, "the importance weights" is the cost weights, and I have made the correction.
>
> **9. L219** Here’s a revised version: We construct an expert dataset and a violating dataset to evaluate Equation 6 offline.
>
> **10. L248** In L249, we add references [1] and [2], which experimentally demonstrate that excessively high rewards can incentivize agents to violate constraints.
>
> **11. L283** We think that expert policy should have the highest possible survival rate to ensure the correctness of their policy. And it is likely that the deaths in the dataset are caused by issues with the doctors' policy, so we hope to exclude these disturbances.
>
> **Response 6:**
>
> (1) Figures 8 and 7 use the same statistical methods. In Figure 8, in the mechanical ventilator environment, the relationship between different algorithm strategies (DDQN, CQL,...) and the action gap with the doctor's strategy (PEEP DIFF and FiO2 DIFF) and mortality rate is illustrated. The horizontal axis represents the action parameters' (PEEP and FiO2) DIFF, while the vertical axis represents the mortality rate.
>
> (2) Lower is better. The lower the proportion of "too high" and "sudden change" the better.
>
> (3) max ∆ represents the maximum change in medication dosage.
>
> References:
>
> [1] Guiliang Liu, et. al. Benchmarking constraint inference in inverse reinforcement learning. 2022.
>
> [2] Zuxin Liu,et. al. Constrained decision transformer for offline safe reinforcement learning. 2023.

---

> > ### Comment · Reviewer_cbZJ · 2024-08-12
> >
> > While I thank the authors for the responses, I am maintaining my assessment of the submission given how substantial the clarifications and revisions are. I hope the authors can incorporate these to strengthen the paper.

---

> > > ### Author Response · Authors · 2024-08-14
> > >
> > > Thank you again for reviewing our work! We will continue to refine the paper. If you have any further questions, we would be happy to discuss them with you.

---

### Official Review · Reviewer_sHtP · 2024-07-12

**Soundness:** 3
**Presentation:** 2
**Contribution:** 3
**Rating:** 5
**Confidence:** 2

**Summary:**

The paper uses the Inverse Constrained Reinforcement Learning (ICRL) framework to infer constraints in healthcare problems from expert demonstrations. It proposes the Constraint Transformer (CT) to address the dependence of decisions on historical data, which is generally ignored in ICRL methods with Markovian assumptions. It borrows the causal transformer from the previous decision transformer to incorporate history into constraint modeling. Additionally, a model-based offline RL model is trained to generate violating data. The CT demonstrates improved safety by effectively modeling constraints based on both violating and expert data.

**Strengths:**

-	The paper addresses a gap in existing ICRL applications by integrating historical data into the decision-making process.
-	The paper augments the violating data in the offline training dataset with a generative world model.
-	The proposed method has been thoroughly evaluated in three aspects: effective constraints, improved sepsis strategies, and safe policies.

**Weaknesses:**

-	The proposed method depends heavily on the generated violating data, which defines the objective function in the constraint transformer. How sensitive is the estimated policy to the generative world model? Figure 12 shows that the action distributions in the expert dataset and the violating dataset are different. The VASO action seldom takes a large value in the violating dataset. Will this distribution difference cause any trouble in the learning of the constraint?
-	Could the authors provide some details on how the DIFF between the estimated policy and the physicians' policy is calculated through graphical analysis? It would also be helpful if the authors could explain how Figure 7 is plotted. Is it calculated based on the dosage differences at each timestamp? In addition, what are the implications of the three DIFF evaluation metrics in Table 2? Since both IV and VASO are part of the action space, is the ACTION DIFF alone sufficient to evaluate the estimated policy?

**Questions:**

See above.

**Limitations:**

The authors have addressed the limitations of their work.

---

> ### Author Rebuttal · Authors · 2024-08-07
>
> Thank you for the thoughtful review of our work! Please allow us to address your concerns and answer the questions.
>
> **Weakness 1:**
>
> **Response:**
>
> **(1)	How sensitive is the estimated policy to the generative world model?**
>
> To explore the sensitivity of the estimated policy to the generative world model, we designed the following experiment. The quality of the data generated by the world model is determined by the target reward. As the target reward increases, the world model generates more aggressive data in order to obtain higher rewards. So we set the target reward to 1, 5, 10, 20, and 30, respectively, and observed the impact of the generated data on the policy, as shown in Table 2. As the target reward increases, the performance of the policy improves; however, there is an upper limit, and it will not increase indefinitely.
>
> **(2)	Will this distribution difference cause any trouble in the learning of the constraint?**
>
> We think that this impact exists, but it will not be significant. Firstly, the generative model used for creating the non-compliant dataset in this paper is a reinforcement learning (RL) model capable of generating data within legal boundaries (see Appendix B.1 in the paper). The actions, states, and rewards it generates must all fall within these legal boundaries. Therefore, this generative model is less likely to produce extremely high drug dosages. However, previous work has confirmed that such models can indeed generate non-compliant data [1]. Consequently, this generative model still presents a "potential risk."
>
> **Weakness 2:**
>
> **Response:**
>
> **(1) Could the authors provide some details on how the DIFF between the estimated policy and the physicians' policy is calculated through graphical analysis?**
>
> In the real medical dataset, we know the patient's state and the drug dosage (a) under the doctor's policy. We use the estimated policy to provide the drug dosage (b) for the same patient state under the estimated policy. We then calculate the DIFF for each patient state, which is $b−a$. We also calculate the mortality rate and standard deviation (std) of patients under the doctor's policy for different DIFF values.
>
> In Figure 7, the x-axis represents the DIFF (the difference in drug dosage between the two policies), and the y-axis represents the mortality rate of all patients under the doctor's policy at that DIFF value. Observing the point where the x-axis is 0, which indicates no difference between the estimated policy and the doctor's policy, we see that when this point has the lowest mortality rate, it suggests that the estimated policy is closer to an ideal policy with a lower mortality rate. This indicates that the estimated policy is a safer policy.
>
> **(2) Is it calculated based on the dosage differences at each timestamp?**
>
> Yes, we calculate the dosage differences for each timestamp.
>
> **(3) In addition, what are the implications of the three DIFF evaluation metrics in Table 2?**
>
> IV DIFF represents the dosage difference of IV medication between the two policies at each timestamp. VASO DIFF represents the dosage difference of VASO medication between the two policies at each timestamp. ACTION DIFF represents the difference between the two policies for the vector composed of IV and VASO medications at each timestamp.
>
> **(4) Since both IV and VASO are part of the action space, is the ACTION DIFF alone sufficient to evaluate the estimated policy?**
>
> ACTION DIFF can evaluate the policy individually. However, this might present a problem: even if we standardize the dosages of the two medications to the same dimension, there may still be cases where the policy performs better for only one specific action. Therefore, for a more comprehensive evaluation, we need to focus on the performance of actions in each dimension.
>
> References:
>
> [1] Zuxin Liu, Zijian Guo, Yihang Yao, Zhepeng Cen, Wenhao Yu, Tingnan Zhang, and Ding 506 Zhao. Constrained decision transformer for offline safe reinforcement learning. arXiv preprint 507 arXiv:2302.07351, 2023.

---

> > ### Comment · Reviewer_sHtP · 2024-08-12
> >
> > Thank you for your thoughtful and detailed response to my comments. I have no further questions at this time.

---

### Official Review · Reviewer_guwJ · 2024-07-12

**Soundness:** 2
**Presentation:** 3
**Contribution:** 2
**Rating:** 5
**Confidence:** 4

**Summary:**

This paper introduces the Constraint Transformer (CT) framework to enhance safe decision-making in healthcare. The proposed CT model uses transformers to incorporate historical patient data into constraint modelling and employs a generative world model to create exploratory data for offline RL training. The authors supported their points by presenting experimental results in scenarios like sepsis treatment, showing that CT effectively reduces unsafe behaviours and approximates lower mortality rates, outperforming existing methods in both safety and interoperability.

**Strengths:**

This paper shows its strengths in the following aspects:
- The paper addresses the novel angle of ensuring safety in offline reinforcement learning (RL) for healthcare, a critical and previously underexplored issue.
- It incorporates Inverse Constrained Reinforcement Learning (ICRL) into offline reinforcement learning (RL) for healthcare, introducing a novel approach to inferring constraints from expert demonstrations in a non-interactive environment.
- The implementation of a causal transformer to learn the constraint function is interesting, allowing the integration of historical patient data and capturing critical states more effectively.
- Extensive results on 2 datasets are presented. The proposed Constraint Transformer (CT) framework is shown to reduce unsafe behaviours and approximates lower mortality rates.

**Weaknesses:**

Despite its strengths and novelty, this paper suffers from several critical technical flaws, primarily concerning the soundness of evaluation rather than the method itself:

1. **Definition of Metric**: The metric $\omega$ is defined by comparing drug dosages related to **mortality rate**, which I believe is a flawed definition, even though it has been used in previous papers. Mortality rate can be influenced by numerous factors, making it unsuitable as a reward for RL, which considers a limited number of drugs. It is challenging to convince clinicians that mortality can indicate the 'treatment quality' of vasopressor or mechanical ventilation. This suggests that the reward is not solely a function of the previous action and state but also many unconsidered features (hidden variables) in the datasets, such as adrenaline, dopamine, historical medical conditions, phenotypes, etc. [1] pointed out that doctors usually set a MAP target (e.g., 65) and administer vasopressors until the patient reaches this safe pressure; [2] suggests using the NEWS2 score as the reward supported by clinical evidence. None of these directly use mortality. While it is understandable that this paper is not a clinical study, and hence, it is not the authors' responsibility to identify clinically appropriate reward designs, I recommend referring to [1] and [2] for a reward design that makes more clinical sense.

2. **Definition of Optimal Policy**: This paper follows [3]'s definition of optimal policy. From my understanding, the clinician's policy $\hat{\pi}$ is approximated by a neural network. [2] pointed out that in the sepsis dataset, the behaviour policy can result in critical flaws in a very small number of states. Although the number of incorrect predictions is limited, they can still bias the off-policy evaluation results severely. I suspect this paper may encounter a similar issue. The authors should provide experiments and visualizations on the learning quality of the behaviour policy to justify their approach.

3. **Model-Based Off-Policy Evaluation**: The data imbalance in both the sepsis and ventilation datasets is significant. It is questionable whether the learned model can generalize well. The most acceptable way to validate the method remains using simulated data where all policies can be tested online. One possible testbed is the DTR-Bench[4] medical simulated environment.

The paper has a few other minor technical flaws compared to the above three.

[1] Jeter, Russell, et al. "Does the" Artificial Intelligence Clinician" learn optimal treatment strategies for sepsis in intensive care?." arXiv preprint arXiv:1902.03271 (2019).

[2] Luo, Zhiyao, et al. "Position: Reinforcement Learning in Dynamic Treatment Regimes Needs Critical Reexamination." Forty-first International Conference on Machine Learning.

[3] Aniruddh Raghu, Matthieu Komorowski, Leo Anthony Celi, Peter Szolovits, and Marzyeh Ghassemi. Continuous state-space models for optimal sepsis treatment: a deep reinforcement learning approach. In Machine Learning for Healthcare Conference, pages 147–163. PMLR, 2017

[4] Luo, Zhiyao, et al. "DTR-Bench: An in silico Environment and Benchmark Platform for Reinforcement Learning Based Dynamic Treatment Regime." arXiv preprint arXiv:2405.18610 (2024).

**Questions:**

Attention Mechanism: The paper highlights the importance of the causal attention mechanism in the Constraint Transformer. Since the two experimental datasets are both short-term time series datasets, is transformer really necessary? Is it possible that an RNN can be better than a transformer?

**Limitations:**

There is no negative societal impact or limitation that needs clarification.
In addition to the weakness I mentioned, this paper:
1. lacks off-policy evaluation results.
2. may summarise more related work in this 'dynamic treatment regime' field. A few examples are listed below:

[1] Kondrup, F., Jiralerspong, T., Lau, E., de Lara, N., Shkrob, J., Tran, M. D., Precup, D., and Basu, S. Towards safe mechanical ventilation treatment using deep offline reinforcement learning. In Proceedings of the AAAI Conference on Artificial Intelligence, volume 37, pp. 15696–
15702, 2023.

[2] Liu, Y., Logan, B., Liu, N., Xu, Z., Tang, J., and Wang, Y. Deep reinforcement learning for dynamic treatment regimes on medical registry data. In 2017 IEEE international conference on healthcare informatics (ICHI), pp. 380–385. IEEE, 2017.

[3] Nambiar, M., Ghosh, S., Ong, P., Chan, Y. E., Bee, Y. M., and Krishnaswamy, P. Deep offline reinforcement learning for real-world treatment optimization applications. In Proceedings of the 29th ACM SIGKDD Conference on Knowledge Discovery and Data Mining, pp. 4673–4684, 2023.

[4] Peng, X., Ding, Y., Wihl, D., Gottesman, O., Komorowski,M., Li-wei, H. L., Ross, A., Faisal, A., and Doshi-Velez,F. Improving sepsis treatment strategies by combining deep and kernel-based reinforcement learning. In AMIA Annual Symposium Proceedings, volume 2018, pp. 887. American Medical Informatics Association, 2018.

---

> ### Author Rebuttal · Authors · 2024-08-06
>
> Thank you for the thoughtful review of our work! Please allow us to address your concerns and answer the questions.
>
> **Weakness 1:** Definition of Metric
>
> **Response:**
>
> **(1) Reward Function Design:** The reviewer may have misunderstood our reward function design. In our work, the reward function did not directly utilize mortality rates but rather included intermediate rewards (as shown in Appendices B.1 and B.2). For example, in the design for sepsis, intermediate rewards like the SOFA score were included. For different diseases, we designed different reward functions based on previous literature. Additionally, we agree with the reviewer's comment that "complex reward design can facilitate the learning of strategies," but the design of reward functions is relatively challenging and requires the involvement of medical experts. We also recognize that the current reward function design may not account for hidden variables (potentially fatal). Therefore, we use a relatively simple reward function, incorporating a cost function with historical dependency, to take into account the changes in indicators like MAP and NEWS that were not considered in the reward function. This will guide the agent in learning safe and effective strategies.
>
> To verify whether our penalty function can capture changes in NEWS and MAP, we conducted supplementary experiments as shown in Figure 1. When the NEWS score is too high, the penalty value increases accordingly; similarly, when MAP is outside the normal range, the penalty value also increases. This indicates that the penalty function can compensate for the shortcomings of the reward function design.
>
> **(2) Evaluation Metrics:** We acknowledge that the evaluation metric ω is not an ideal measure. Therefore, we also consider the relationship between the penalty value and medical metrics (section 5.1) and the probability of dangerous actions in the policy (section 5.2) as part of the evaluation criteria to compare the safety of different policies from multiple perspectives. In the experiments, all methods are based on the same reward function, so the design of the reward function is not a variable factor.
>
> **Weakness 2:** Definition of Optimal Policy
>
> **Response:**
>
> **(1) Behavior Policy Fitting Error:** In our Offline RL, we did not fit the clinicians' policy using a neural network, so there is no fitting error in this part of the experiment. The reviewer has provided a valuable suggestion, highlighting that fitting the behavioral policy could indeed help eliminate some confounding factors. In our supplementary experiments, we utilized neural networks to fit the behavioral policy, corrected the model accordingly, and then used OPE evaluation to conduct the offline policy evaluation experiment.
>
> **(2) Offline Policy Evaluation:** We supplemented our work by referring to [1] for the offline policy evaluation as follows: Using the same behavioral fitting function and the same NEWS2 reward, we compared the results of the policy under different evaluation metrics, as shown in Table 1. The CDT+CT method performed better than other methods on the RMSE_IV, WIS, WIS_b, and WIS_bt evaluation metrics.
>
> **Weakness 3:** Model-Based Off-Policy Evaluation
>
> **Response:** Since we aim to evaluate whether there are instances of excessively high or sudden changes in medication dosage, our model's action space consists of the actual dosage values. However, DTR-Bench currently cannot perform online evaluations of medication dosages, as it only provides a discrete action space (i.e., "yes" or "no" for administering medication). In this case, there is no issue of overestimation, so this online testing method cannot assess whether the policy effectively avoids dangerous behaviors. Exploring a simulation environment based on continuous action spaces is a promising direction for future research.
>
> **Question:** Attention Mechanism: The paper highlights the importance of the causal attention mechanism in the Constraint Transformer. Since the two experimental datasets are both short-term time series datasets, is transformer really necessary? Is it possible that an RNN can be better than a transformer?
>
> **Response:** Compared to RNNs, the Transformer architecture has the following advantages:
>
> **(1) Computational Efficiency:** Due to the global computation allowed by the Transformer's self-attention mechanism, it can be highly parallelized during training[2,3], leading to faster training speeds. Transformers can significantly improve training efficiency. RNNs, on the other hand, require sequential processing of each time step, which makes parallelization difficult and results in slower training speeds.
>
> **(2) Capturing Complex Information:** Transformers utilize multi-head attention mechanisms to simultaneously focus on different parts of the sequence, allowing them to better capture complex relationships in medical data. In medical datasets, events might be recorded with non-uniform time intervals. The Transformer's self-attention mechanism does not rely on fixed time steps, making it more flexible in handling such situations.
>
> **(3) Interpretability:** The self-attention mechanism of Transformers makes it easier to understand which parts of the data sequence the model is focusing on, providing better interpretability, which is crucial in the medical field. RNNs, with their internal states and memory units, are more difficult to interpret, which may reduce the transparency of clinical decision-making.
>
> References:
>
> [1] Luo, Zhiyao, et al. "Position: Reinforcement Learning in Dynamic Treatment Regimes Needs Critical Reexamination." Forty-first International Conference on Machine Learning.
>
> [2] Peng B, Alcaide E, Anthony Q, et al. Rwkv: Reinventing rnns for the transformer era[J]. arXiv preprint arXiv:2305.13048, 2023.
>
> [3] Yi Tay, Mostafa Dehghani, Dara Bahri, and Donald Metzler. 2022. Efficient transformers: A survey. ACM Computing Surveys, 55(6):1–28.

---

> ### Comment · Reviewer_guwJ · 2024-08-08
> **First reply to the rebuttal**
>
> Thanks to the authors for providing their views and justification.
>
> Regarding weakness 1:
> I agree with your explanation to reward design and I appreciate the added cost function for NEWS and MAP. However, I still have 2 concerns:
> 1. You did not compare naive baselines (random, zero-drug, etc.) in your added OPE result; according to [1], including naive baselines are critical. You may also add naive baselines to other experiments.
> 2. The OPE result is generally good. RMSE_VASO is significantly larger than offline RL algos while RMSE_IV is not. You might visualise the policy and check if the difference is caused by a reduced use of VASO, therefore preventing high dose.
> 3. No OPE result on ventilation dataset
>
> ## Regarding weakness 3:
> Yes, I agree with the authors that 'dangerous' actions (according to your definition) are actual values that do not need a model for prediction. However, I think the authors have some confusion about the concept of 'dangerous actions' and 'dangerous states'.
>
> Technically, 'dangerous actions' don't mean a high dose, a high dose doesn't always lead to dangerous states. It does not seem right to define a high dose to be dangerous, as some high doses might be necessary. Even if the agent proposed high doses while the doctors did not, it does not mean the agent is acting less dangerously or less optimally, as we do not have counterfactuals without simulation environments. Combined with the sad fact that all your evaluations are offline on imbalanced datasets, it is tough to convince me that your approach is 'safer'.
>
> Let me give you a very simple example to show that low doses can be dangerous: the recommended dose of insulin for Type 1 diabetic patients is 0.5 units/day, roughly. Policy 1 gives 0.25 units at noon and night and 0 elsewhere; policy 2 gives 0.01 units every 1 min for 24 hours. Accumulatively, the policy 2 will kill the patient because the summed dosage is too high. However, we did not observe any single 'high' dosage in policy 2. This example shows that we cannot simply say high-dose action is dangerous. There are a lot of complicated PK/PD effects we might consider further beyond high doses. I recommended using online testing because the simulation environment will consider the PK/PD for you, such that a higher reward means a 'safer' performance. If you disagree with this please justify further.
>
> "DTR-Bench currently cannot perform online evaluations of medication dosages, as it only provides a discrete action space"
> No, DTR-Bench has four environments, three of which have continuous action spaces and the sepsis one has a discrete action space. The sepsis environment's states are human-readable (because they are designed in that way). By checking the state index, you can tell how many vital signs are abnormal. The authors should read more literature on the dynamic treatment regime field.
>
> Also, I am confused by the authors' statement that 'there is no overestimation'. From my understanding, all value-based or actor-critic methods have overestimation issues. The authors may explain further.
>
> In conclusion, the justification for weakness 3 is weak.
>
> ## Regarding the question:
> The authors might answer my previous questions directly: "Since the two experimental datasets are both short-term time series datasets, is a transformer really necessary? Is it possible that an RNN can be better than a transformer?"
> I asked this because many exciting architectures or models in other fields do not work in healthcare. There are numerous reasons, but one is that healthcare data is imbalanced and insufficient for large models to train. Some works in healthcare show that transformers do not improve performance in small-scale healthcare tasks; some even decrease due to over-parameterization and/or overfitting. I would invite the authors to eliminate my concern through ablation studies. For example, what if you replace the transformer with a shallow encoder-decoder LSTM?
>
> Apart from the weaknesses I mentioned before, I have a few minor questions that were not presented. The authors might answer if time allows:
> 1. Is it possible that a human-defined cost function can surpass some baselines or your method? For example, with the help of LLMs, a non-healthcare expert can know the normal range of all clinical variables and define a heuristic cost function very easily. Since this knowledge is prior, it should intuitively outperform most baselines and possibly your method. If so, is the motivation of your paper solid?
> 2. Do you have any assumptions for the optimality of the behavioural policy? Will that affect the choice of your baselines?
>
> The current rebuttal are insufficient to allow me change the score. I look forward to further discussion if anything mentioned above is not agreeable or wrong.
>
>
> References:
>
> [1] Luo, Zhiyao, et al. "Position: Reinforcement Learning in Dynamic Treatment Regimes Needs Critical Reexamination." Forty-first International Conference on Machine Learning.

---

> > ### Author Response · Authors · 2024-08-12
> >
> > Thank you for the thoughtful comments on our work! Please allow us to address your concerns and answer the questions.
> >
> > ## Weekness 1
> >
> > ## Response:
> >
> > 1. Thank you for your suggestions. We will add naive baselines to our experiments.
> >
> > 2. This is indeed a very good approach to observe the differences in actions. In our study, the action space ranges from 0 to 24. Our experiments revealed that the agent rarely takes actions where both drug doses are at the maximum level (i.e., 24), while there is no significant reduction in other actions. However, the agent tends to choose higher VASO drug doses compared to doctors. We are considering whether the discrete action space has influenced this result, leading the model to perceive that danger only arises when both drug doses are high.
> >
> > 3. Sorry, we have not yet completed this part of the experiment, but we will continue to supplement this aspect in our subsequent research.
> >
> > ## Weakness 3
> >
> > ## Response:
> >
> > Our previous statement may have been unclear. "Too high" refers to a drug dosage that is lethal for all patients. This was just an example of a dangerous action, and there could also be the type of dangerous action you mentioned, where the total dosage is too high. However, our model does not restrict the categories of "dangerous actions." In our paper, we tested "too high" and "sudden change." However, we have not tested whether the model can restrict other types of "dangerous actions." We plan to test this in other continuous action space environments within the DTR Bench in the future.
> >
> > By "not overestimating," we mean that in the sepsis environment of the DTR Bench, the actions taken are "yes" or "no." This action space may make it difficult to determine whether a drug dosage is high or low. It is not an issue of overestimation based on value or actor-critic methods.
> >
> > ## Question
> >
> > ## Response:
> >
> > Thank you for your response. I understand your concerns. We apologize that we have not yet completed the LSTM ablation experiments. To improve our work, we will conduct additional experiments in the future to enhance our study.
> >
> > ## Other questions
> >
> > ## Response:
> >
> > 1. Regarding the cost function defined by humans, we designed one based on expert opinions in the paper. Designing a cost function by humans can be challenging, particularly in determining hyperparameters. With the help of LLMs, we consider that there may be some latent state indicators that are currently unknown in the medical field. Therefore, this prior knowledge may be difficult to provide to large models.
> >
> > 2. The optimality of behavioral strategies is indeed hard to define. Since a zero-mortality action strategy does not exist—because some patients will die regardless of treatment—we assume that a strategy with a lower mortality rate is more optimal. For the baseline selection, we chose the expert strategy (i.e., the physician's strategy data with a patient mortality rate of 0) and the zero drug dosage strategy.
> >
> > Thank you for your response. We will continue to supplement our work with the following experiments: other environment OPE tests, DTR Bench online testing, and LSTM ablation experiments, to further strengthen our study.

---

> > > ### Comment · Reviewer_guwJ · 2024-08-13
> > >
> > > According to the current update of the authors results, I believe the score can be increased to 5. I'm happy to discuss further with AC and other reviewers

---

### Official Review · Reviewer_R5ke · 2024-07-13

**Soundness:** 2
**Presentation:** 2
**Contribution:** 3
**Rating:** 5
**Confidence:** 2

**Summary:**

The authors consider healthcare applications of RL algorithms in which implicit constraint modeling is critical for safe recommendations.
This is modeled as an RL policy optimization with constraints. However, the constraints are often unknown and need to be inferred from expert data trajectories in the healthcare applications. The authors propose a neural network estimator to combine with the constrained MDP formulation. They also identify that the naive way to represent states leads to non-Markov structure. To address these issues, the paper proposes a simple modification (based on prior work on the "preference attention layer") to a causal transformer based policy which attempts to model a parametric constraint function. Evaluations are based on healthcare benchmarks.

**Strengths:**

- The paper considers a practically motivated approach grounded in real data for a healthcare application where data could be challenging to obtain.
- Proposes simple addition to the final layer of a causal transformer to parametrize constraints on trajectories learned from expert data.
- Evaluations are conducted on real healthcare domain benchmarks and extensive ablations are included to ensure that the architecture change is meaningful in obtaining the improvements.

**Weaknesses:**

- It seems like the system (which is the patient) is not feasible to model as evolving according to an Markov process on the observed state at each time, but instead a POMDP with a high dimensional latent state.

- Clarity and presentation can be improved. In Equation (4), what is $\hat{\tau}$, this was never defined for being such a key component of the procedure. Similar issues in Eq (8) related to clarity.

**Questions:**

- In explaining the main Equation (4), the blurb in lines 162-164 needs some attention. Specifically, what does "MDP obtained after augmenting M with cost function $C_\theta$, using the executing policy ..." mean exactly?

- In Eq (8) what does $s_t, r_{t-1} \in x_t \sim D_e$ represent? It looks like $x_t$ is a representation generated by the transformer, and $s_t, r_t$ are from the data. Is there a problem in the notation?

- Is the transformer representations being trained from the gradients of both the policy model as well as the world model? If so, how do they interact? If not, consider adding stop gradients at the relevant places in Eq (7) and/or (8) to make this explicit.

**Limitations:**

Yes.

---

> ### Author Rebuttal · Authors · 2024-08-06
>
> Thank you for the thoughtful review of our work! Please allow us to address your concerns and answer the questions.
>
> **Weakness 1:** It seems like the system (which is the patient) is not feasible to model as evolving according to an Markov process on the observed state at each time, but instead a POMDP with a high dimensional latent state.
>
> **Response:** Thank you for your suggestion. We agree that the complexity of the patient system should be described as a POMDP, as the MDP model indeed oversimplifies the system. We constructed a non-Markovian model by incorporating historical state sequences into the RL decision-making process, with the goal of capturing potential states from history, which aligns with your suggestion. To ensure a more rigorous presentation, we have reformulated it as a CPOMDP framework in Section 3 Problem Formulation, defined as $\left( \mathcal{S},\mathcal{A},\mathcal{O},\mathcal{P},\mathcal{Z},\mathcal{R},\mathcal{C},\gamma ,\kappa,\rho_0 \right)$, where $\mathcal{O}$ is the set of observations $o$, $\mathcal{P}\left( s'|s,a \right) =Pr\left( s_{t+1}=s'|s_t=s,a_t=a \right) $ the transition probability, and $\mathcal{Z}\left( o|s',a \right) =Pr\left( o_{t+1}=o|s_{t+1}=s',a_t=a \right)$  is the observation probability. The other parameters have been explained in the paper.
>
> **Weakness 2:** Clarity and presentation can be improved. In Equation (4), what is $\hat{\tau}$, this was never defined for being such a key component of the procedure. Similar issues in Eq (8) related to clarity.
>
> **Response:** We apologize for any inconvenience caused in your reading. The explanations for the two points mentioned above are as follows:
>
> 1. In Equation (4), $\hat{\tau}$ is the trajectory which is sampled from the executing policy (training policy) $\mathcal{M}^{\hat{\zeta}_{\theta}}$. In an online environment, we can use the execution policy to interact with the environment and generate some trajectories $\hat{\tau}$.
>
> 2. In Equation (8),  $x_{t}=\{h_{t} \cup a_{t}\}= \( R_{-K: t}, s_{-K: t}, a_{-K: t} \)$ is the input which includes the reward $R$, states $s$ and action $a$ from the preceding $K$ timesteps, where $K$ is the context length. The input is encoded by linear layers and passes through the casual transformer to predict hidden tokens $g_t$. Next, we use the hidden tokens as input and employ two linear layers ($\ell_{\mu} \text{ and } \ell_{\varphi}$) to predict the current reward $r_{t-1}$ and the next state $s_t$, with the objective of minimizing the mean squared error for each linear layer, defined as: $
>     \min_{\varphi,\mu} \mathbb{E} [{(s_t-\ell_{\varphi}(g_t))^2} + {(r_{t-1}-\ell_{\mu}(g_t))^2}] $
>
> In addition, we will improve the overall clarity and presentation of the paper to avoid such issues.
>
> **Question 1:** In explaining the main Equation (4), the blurb in lines 162-164 needs some attention. Specifically, what does "MDP obtained after augmenting M with cost function $C_{\theta}$, using the executing policy ..." mean exactly?
>
> **Response:** The ICRL approach involves two policies: one is the expert policy $\pi_e$, and the other is the policy being trained (execution policy $\pi_{ \mathcal{M}^{\zeta_{\theta}}})$. This sentence defines the Markov Decision Process (MDP) model for the latter. $\mathcal{M}^{\zeta_{\theta}}$ represents the MDP that results from adding the cost function $C_{\theta}$ to the original MDP M. The executing policy for this augmented MDP is denoted as $\pi_{\mathcal{M}^{\zeta_{\theta}}}$.
>
>
> **Question 2:** In Eq (8) what does $s_t,r_{t-1} \in x_t \sim \mathcal{D}_e$ represent? It looks like $x_t$ is a representation generated by the transformer, and $s_t,r_t$ are from the data. Is there a problem in the notation?
>
> **Response:** Sorry, there is indeed an issue here. We have redefined the hidden tokens generated by the transformer as $g_t$.
>
> **Question 3:** Is the transformer representations being trained from the gradients of both the policy model as well as the world model? If so, how do they interact? If not, consider adding stop gradients at the relevant places in Eq (7) and/or (8) to make this explicit.
>
> **Response:** Yes, in the model-based offline RL framework, the transformer is trained together with the policy model and the world model. The objectives of the policy model and the world model are to generate actions, rewards, and the next state, respectively. The transformer structure is used to extract historical information for the policy and world models. When training model-based offline RL, the goal is to simultaneously minimize Equations (7) and (8). During this process, the transformer is also trained alongside the objectives until convergence.

---

> > ### Comment · Reviewer_R5ke · 2024-08-11
> > **Acknowledgement of rebuttal**
> >
> > Thank you for the clarifications. I believe my original review scores are fair and stick to them.

---

### Author Rebuttal · Authors · 2024-08-07

We thank the reviewers for their time, suggestions and questions that we believe will improve the quality of the paper. Below we summarize our overall response to the reviewer’s questions and comments.

- We will add a discussion on the relationship between the key metrics, NEWS and MAP, and the cost function, as shown in Figure 1 of the attachment. The cost function can indeed capture dangerous states that the reward function overlooks, addressing the issue of latent variables that cannot be incorporated into the reward function.

- We will include an off-policy evaluation as suggested by reviewer guwJ. Referring to [1], we will evaluate our method and others using multiple evaluation metrics under the same reward function and behavior policy, as shown in Table 1 of the attachment. Our policy performs well in $RMSE_{IV}$、$WIS_b$、$WIS_t$、$WIS_{bt}$. For the specific meanings of these metrics, please refer to [1].

- We will add a sensitivity analysis of the generative world model to CDT+CT, as suggested by reviewer sHtP. As the target reward increases, the generated world model exhibits more aggressive behavior, which can improve the performance of the estimated policy, but there is an upper limit to this effect.

- We will clarify the meaning and calculation process of the evaluation metric
$\omega$, as suggested by reviewers sHtP and cbZJ. For details, please refer to our responses to reviewer sHtP (Response2-1) and cbZJ (Response5-3).

References:

[1] Luo, Zhiyao, et al. "Position: Reinforcement Learning in Dynamic Treatment Regimes Needs Critical Reexamination." Forty-first International Conference on Machine Learning.

---

### Decision · Program_Chairs · 2024-09-25

**Decision:**

Reject

**Comment:**

The paper proposes to build on prior work on inverse constrained reinforcement learning to enable safe decision-making addressing two main challenges: i) using a causal transformer to enable non-markovian state-space and subsequently policy function, ii) using a generative world model that enables data-augmentation using offline healthcare data. The overall constrained MDP solution is empirically evaluated on healthcare data to generate safe policies, data-augmentation using the world model, and modeling constraints.

Reviewers consider the problem formulation well grounded in healthcare and provided meaningful empirical evaluation setup. However there are several presentation and clarity issues that are affecting the quality of the contributions. Overall I agree with some lack of clarity in setup, notation, as well as some claims made such as those pointed out by all reviewers, especially by cbZJ that state definitions could be redefined based on what is being captured. Further the choice of metric and reliance on mortality were also concerns raised for the specific motivation. Overall, the authors have addressed many of the raised questions, like considering ablations with respect to learned behavior policy, and agreed to address the typos and other framework and writing issues. However, considering the claimed benefit, fully addressing all issues, including the need for sensitivity analysis, and off-policy evaluation of the proposed method's output, clarifying the choice of evaluation metrics will require significant changes and warrants another round of peer review.

Therefore, I encourage the authors to continue to improve their work, and recommend a rejection for NeurIPS